# Organic coating on sulfate and soot particles in summer Arctic atmosphere

Hua Yu[1,2], Weijun Li[2*], Yangmei Zhang[3], Peter Tunved[4], Manuel Dall'Osto[5], Xiaojing Shen[3], Junying Sun[3], Xiaoye Zhang[3], Jianchao Zhang[6], Zongbo Shi[7,8*]

[1]College of Life and Environmental Sciences, Hangzhou Normal University, 310036, Hangzhou, China

[2]Department of Atmospheric Sciences, School of Earth Sciences, Zhejiang University, 310027, Hangzhou, China

[3]Key Laboratory of Atmospheric Chemistry, Chinese Academy of Meteorological Sciences, Beijing, China

[4]Department of Environmental Science and Analytical Chemistry, Stockholm University, 10691, Stockholm, Sweden

[5]Institute of Marine Sciences, ICM-CSIC, Passeig Mar fim de la Barceloneta, 37-49, E-08003, Barcelona, Spain

[6]Key Laboratory of the Earth's Deep Interior, Institute of Geology and Geophysics, Chinese Academy of Sciences, 100029, China

[7]School of Geography, Earth and Environmental Sciences, University of Birmingham, Birmingham, UK

[8]Institute of Surface Earth System Science, Tianjin University, Tianjin, China

[*]Corresponding Emails: liweijun@zju.edu.cn; z.shi@bham.ac.uk

**Abstract**

Interaction of anthropogenic particles with radiation and clouds plays an important role on Arctic climate change. Mixing state of aerosols is a key parameter to influence aerosol-cloud and aerosol-radiation interaction. However, little is known on this parameter in the Arctic, preventing an accurate representation of this information in global models. Here we used transmission electron microscopy with energy-dispersive X-ray spectrometry (TEM/EDS), scanning TEM, scanning electron microscopy (SEM), nanoscale secondary ion mass spectrometry (NanoSIMS), and atomic forces microscopy (AFM) to determine the size and mixing properties of individual particles at 100 nm – 10 μm, with a particular focus on sulfate and carbonaceous particles. We found that non-sea salt sulfate particles with size range at 100-2000 nm were commonly coated with organic matter (OM) in summer. 20% of sulfate particles also had soot inclusions which only appeared in the OM coating. The OM coating is estimated to contribute to 63% of the particle volume on average. To understand how OM coating influences optical properties of sulfate particles, the Mie theory of the core-shell model was applied to calculate optical properties of individual sulfate particles. The result shows that absorption cross section (ACS) of individual OM-coated particles significantly increased when assuming the OM coating as light-absorbing brown carbon (BrC) and the ACS also increased following the increasing particle size. The microscopic observations suggest that OM modulates the mixing structure of fine Arctic sulfate particles, which may determine their hygroscopicity and optical properties.

## 1. Introduction

Surface temperatures are rising faster in the Arctic than the rest of globe (IPCC, 2013). Although increased human-induced emissions of long-lived greenhouse gases are certainly one of the driving factors, air pollutants, such as aerosols and ozone, are also important contributors to climate change in the Arctic (Law and Stohl, 2007; Shindell, 2007). It is well known that aerosols from northern mid-altitude continents affect the sea ice albedo by altering the heat balance of the atmosphere and surface (Hansen and Nazarenko, 2004; Jacob et al., 2010; Shindell, 2007). These aerosols in Arctic atmosphere include sea salt, sulfate, particulate organic matter (OM), and to a lesser extent, ammonium, nitrate, black carbon (BC) (Hara et al., 2003; Quinn et al., 2007) and mineral dust particles (Dagsson-Waldhauserova et al., 2013). Studies show BC in the Arctic absorbs solar radiation in the atmosphere and when deposited on snow (Iziomon et al., 2006; Koch and Hansen, 2005; Sand et al., 2013; Shindell, 2007). Moreover, Maahn et al. (2017) used aircraft in situ observation of clouds and aerosols and found that concentration of BC are enhanced below the clouds in the Arctic and further influence the mean effective radii of cloud droplets which lead to the suppressed drizzle production and precipitation.

BC, commonly called ''soot'' is derived from the combustion sources such as diesel engines, residential solid fuel, and open burning (Bond et al., 2013). Some studies investigated the possible sources of these BC particles, including natural gas flaring (Qi et al., 2017) and ship emissions in the Arctic (Browse et al., 2013; Weinbruch et al., 2012) and emissions of biomass burning and fossil fuels in the northern hemisphere (Winiger et al., 2016; Xu et al., 2017). For example, Winiger et al.(2017) showed that most Arctic BC is sourced from domestic activities (35%) and transportation (38%), with only minor contributions from gas flaring (6%), power plants (9%), and open fires (12%).

Accumulation of secondary organic aerosols, a significant fraction of the new particles grows to sizes that are active in cloud droplet formation in the Arctic (Abbatt et al., 2019). More than 100 organic species have been detected in the Arctic aerosols and polyacids are the most abundant compound class, followed by phthalates,

aromatic acids, fatty acids, fatty alcohols, sugars/sugar alcohols, and n-alkanes (Fu et
al., 2008). Recently, certain organic aerosols, referred to as brown carbon (BrC), have
been recognized as an important light-absorbing carbonaceous aerosol after BC in the
troposphere (Alexander et al., 2008; Andreae and Gelencser, 2006; Feng et al., 2013;
Lack et al., 2012). BrC can be directly emitted from combustion sources or formed in
the atmosphere via photo-chemical aging (Jiang et al., 2019; Saleh et al., 2013;
Updyke et al., 2012). Moreover, aging of secondary organic aerosols can significantly
contribute to BrC during atmospheric transports (Laskin et al., 2015). Feng et al.(2013)
estimated that on average, BrC accounts for 66% of total OM mass globally and its
light absorption is about 26% of BC.
BC and BrC are often internally mixed with other non-absorbing aerosols, such as
sulfate (Lack et al., 2012; Laskin et al., 2015). Internal mixing means that a single
particle simultaneously contains two or more types of aerosol components (Li et al.,
2016). This internal mixing can enhance BC absorption by a factor of up to two (Bond
et al., 2013) and change the activity of cloud condensation nuclei (CCN) in the Arctic
atmosphere (Leck and Svensson, 2015; Martin et al., 2011). Spatial and temporal
variations of aerosol composition, size distribution, and sources of Arctic aerosols
have been studied extensively in numerous ground-based, ship, airborne observations,
and various atmospheric models (Brock et al., 2011; Burkart et al., 2017; Chang et al.,
2011; Dall Osto et al., 2017; Fu et al., 2008; Hara et al., 2003; Hegg et al., 2010;
Iziomon et al., 2006; Karl et al., 2013; Lathem et al., 2013; Leck and Bigg, 2008;
Leck and Svensson, 2015; Moore et al., 2011; Raatikainen et al., 2015;
Wöhrnschimmel et al., 2013; Winiger et al., 2017; Yang et al., 2018; Zangrando et al.,
2013). A few previous studies also looked at the mixing states of coarse aerosol
particles in Arctic troposphere (Behrenfeldt et al., 2008; Chi et al., 2015; Geng et al.,
2010; Hara et al., 2003; Leck and Svensson, 2015; Moroni et al., 2017; Raatikainen et
al., 2015; Sierau et al., 2014), but those of fine non-sea salt particles, including the
most important short-lived climate forcers – BC and BrC (Feng et al., 2013; Fu et al.,
2008; Kirpes et al., 2018; Laskin et al., 2015; Leck and Svensson, 2015), are poorly
characterized. The poor understanding on mixing state of BC and BrC in individual
particles will prevent the further simulation of atmospheric climate and aerosol-cloud
interaction in the Arctic through the current atmospheric models (Browse et al., 2013;
Samset et al., 2014; Zanatta et al., 2018).

110         In this study, individual aerosol particles were collected in the Arctic during 7-23

August, 2012. We combined the data from various microscopic instruments to
systematically determine the size, composition, and mixing properties of individual
particles, with a particular focus on sulfate and carbonaceous particles. Mie theory
was used to test how OM coating influences optical properties of sulfate particles in
the Arctic when OM was assumed as BrC. The results are discussed in the context of
aerosol-radiation and cloud interaction.

**2. Experimental section**
**2.1 Field campaign**

120         The Svalbard archipelago includes all landmasses between 74 and 81 degrees

North and 10 and 35 degrees East (Figure 1). The islands cover 63000 $km^2$.
Ny-Ålesund town is situated on the west coast of the largest island, Spitsbergen.
Ny-Ålesund town is situated only 1200 km from the North Pole and represents a
central platform for Arctic research. The sampling place represents remote Arctic
conditions.

126         An individual particle sampler at Chinese Arctic Yellow River Station (78°55′N,

11°56′E) collected individual particles (Chi et al., 2015; Geng et al., 2010). The
sampling site is about 2 km far away from the Zeppelin observatory station (78.9N
11.88E) running by the Ny-Ålesund Science Managers Committee
(https://www.esrl.noaa.gov/psd/iasoa/stations/nyalesund). Two to three samples were
regularly collected at 9:00, 16:00, 21:00 (local time) of each day, with a total of 46
samples during 7-23 August, 2012.

133         A sampler containing a single-stage impactor with a 0.5-mm-diameter jet nozzle

(Genstar Electronic Technology, China) was used to collect individual particles by the
air flow rate at 1.5 l min$^{-1}$. Aerosol particles were collected onto copper TEM grids
coated with carbon film. This sampler has a collection efficiency of 31% at 100 nm
aerodynamic diameter and 50% at 200 nm if the density of the particles is 2 g cm$^{-3}$.
The sampler can collect particles with < 10 μm aerodynamic diameter on TEM grids.
Sampling times varied from twenty minutes to two hours in clean remote Arctic area.
After collection, each sample was placed in a sealed dry plastic tube and stored in a
desiccator at 20 ± 3% RH for analysis. Ambient laboratory conditions (17−23% RH
and 19−21 °C) is effective at preserving individual hygroscopic aerosol particles and
reducing changes that would alter samples and subsequent data interpretation
(Laskina et al., 2015). The sample information such as local sampling date and time
and meteorological conditions (e.g., temperature (T), relative humidity (RH), pressure
(P), wind direction (WD), wind speed (WS)) are listed in Table S1.

**2.2 TEM measurement**

149       Individual particle samples were examined by a JEOL JEM-2100 transmission

electron microscopy operated at 200 kV with an energy-dispersive X-ray
spectrometry (TEM/EDS). TEM can observe the mixing structure of different aerosol
components within an individual particle on the substrate because electron beam
transmit through the specimen to form an image. EDS spectra were acquired for a
maximum time of 30 s to minimize potential beam damage and collect particle X-ray
spectra with sufficient intensity. TEM grids are made of copper (Cu) and covered by a
carbon-reinforced substrate, so Cu is excluded from the quantitative analyses of the
particles. Because of the substrate contribution, C content in TEM grid coated by
carbon film might be overestimated in EDS spectra of individual particles.

159       The distribution of aerosol particles on TEM grids was not uniform, with coarser

particles occurring near the center and finer particles on the periphery. Therefore, to
ensure that the analyzed particles are representative, five areas were chosen from the
center and periphery of the sampling spot on each grid. Through a labor-intensive
operation, 2002 aerosol particles with diameter < 10 μm in 21 samples were analyzed
by TEM/EDS (Table S1). To check elemental composition of individual particles,
EDX was manually used to obtain EDS spectra of individual particles. In the clean
Arctic air, there are simply particle types including sea salt, sulfate, soot, and OM.
Because soot particles have chain-like aggregation, it is not necessary to check their
elemental composition. Sea salt particles display spherical or square shapes and are
stable under the electron beam in TEM but sulfate particles are spherical but flats on
the substrate and produce unstable bubble under the electron beam (Buseck and Posfai,
1999; Chi et al., 2015). TEM observations also can clearly identify sulfate particles or
sulfate with OM coating. Therefore, we can easily identify Arctic particle types based
on their morphology. Because of the time-consuming in the experiment, it is not
necessary to frequently check elemental composition of the same particle type. For
the data statistic in this study, we randomly checked elemental composition of 20-30
particles in each sample (Table S1). EDS spectra of 575 particles were manually
selected and saved in the computer for elemental composition analysis. Particles
examined by TEM were dry at the time of observation in the vacuum of the electron
microscope. In our study, the effects of water and other semi-volatile organics were
not considered as they evaporate in the vacuum.
Elemental mapping and line profile of individual aerosol particles were obtained
from the EDX scanning operation mode of TEM (STEM). The STEM information can
clearly display elemental distribution in the targeted individual particles which cannot
be provided by the above EDS examination. Based on preliminary individual analysis,
we further chose the typical samples containing abundant sulfate with OM coating for
the STEM analysis. The high-resolution details of elemental distribution in individual
particles can further prove the details of the mixing structure of sulfate and OM in
individual particles.
The iTEM software (Olympus soft imaging solutions GmbH, Germany) is an
image analysis platform for electron microscopy. In this study, it was used to
manually or automatically obtain area, perimeter, and equivalent circle diameter
(ECD) of individual particles through identifying boundary of every particle in TEM
images. In these analyzed samples, we found there were abundant fine sulfate
particles in 11 samples collected during 9-15 August, 2012. In other samples, there
were only a few sulfate particles and more sea salt particles. Based on the TEM
observations, we selected the samples containing more sulfate particles to further do
other microscopic analyses as below.

### 2.3 NanoSIMS measurement

Because the sulfate particles collected in the Arctic had good consistent property
(e.g., elemental composition and mixing state) from TEM observations, we just
selected three samples containing abundant fine sulfate particles (Table S1) for further
studies. These three samples listed in Table S1 were analyzed using a nanoscale
secondary ion mass spectrometry (NanoSIMS) 50L (CAMECA Instruments,
Geneviers, France) instrument. A micro-cesium source was used to generate $Cs^+$
primary ions, with an impact energy of 16 kV for sample interrogation. The primary
beam was stepped across the sample to produce element specific, quantitative digital
images. The $Cs^+$ primary ion beam was used to obtain $^{16}O^-$, $^{12}C^{14}N^-$, $^{14}N^{16}O^-$, $^{32}S^-$,
$^{35}Cl^-$, and $^{16}O^{23}Na^-$ ions in this study. The NanoSIMS analysis can obtain ion mapping
of particles with nanometer spatial resolution over a broad range of particle sizes
(Figure S1). Because the substrate of TEM grid is carbon, $CN^-$ is adopted to represent
OM in individual particles (Chi et al., 2015; Ghosal et al., 2014). $S^-$ is used to infer the
presence of sulfates in individual particles (Li et al., 2017). Finally, the NanoSIMS
obtained ion mapping of 32 sulfate particles.

### 2.4 SEM and AFM measurement

Because TEM could not vertically observe OM coating and sulfate core, we
conducted one special experiment using a Zeiss ultra 55 scanning electron microscopy
(SEM) with EDS. The TEM grids were mounted onto an aluminum SEM stub and
directly observed in secondary electron image mode. SEM analysis was operated at
10 kv of extra high tension (EHT) and 9.7 mm of work distance (WD). Processes such
as sample moving, analysis region selection and imaging were controlled by computer.
The specimen stage in SEM was tilted at the range of 0-75$^o$, and then we vertically
observed thickness of OM coating and sulfate core on the substrate. To verify vertical
property of individual S-rich particles impacting on the substrate, we observed two
typical samples containing abundant sulfate particles using the SEM (Table S1).
AFM with a digital nanoscope IIIa instrument operating in the tapping mode was
used to observe surface morphology of individual aerosol particles and measure
particle thickness. The tapping AFM has a cantilever and conical tip of 10 nm radius.
By using AFM, a general image of the particles is taken at 10 μm full scan size, which
generally includes 1-2 particles depending on the exact location. In this study, we are
only interested in the sulfate-containing particles. AFM provides surface information
and morphology of 17 particles but no composition. Samples were firstly quickly
examined by the TEM under low magnification mode. In case, the operation roughly
identified S-containing particles and didn't damage the secondary sulfate particles
under the electron. Because TEM grids have coordinates letters, we can exactly find
the same particles on the substrate in AFM examined in TEM experiments. The
procedures can exclude sea salt particles in the AFM image. As a result, the same
samples observed by TEM were then examined in AFM to obtain 3-D image of
secondary sulfate particles and their volume. Because individual particles collected in
Arctic air were scattered on the substrate, we only obtained 17 effective data. After we
obtained AFM images of sulfate particles, the NanoScope analysis software can
automatically obtain bearing area (A) and bearing volume (V) of each analyzed
particle according to the following formula.
$$A = \frac{4}{3}\pi r^2 = \frac{\pi d^2}{3} \rightarrow d = \sqrt{\frac{3A}{\pi}} \qquad (1)$$
$$V = \frac{4}{3}\pi r^3 = \frac{4}{3} \times \frac{\pi D^3}{8} \rightarrow D = \sqrt[3]{\frac{6V}{\pi}} \qquad (2)$$
Where $x$ is the equivalent circle diameter (ECD) and $y$ is the equivalent spherical
diameter (ESD).
ECD of individual aerosol particles measured from the iTEM software can be
further converted into ESD. Based on these data, we estimate one good linear
correlation (y=0.38x) between ESD and ECD of sulfate particles impacting on the
substrate. The value was further used to correct all the analyzed particles in TEM
images (Chi et al., 2015).

**2.5 Calculation of BrC optical properties**

The refractive index used for the non-light-absorbing sulfate component was set to m=1.55 at 550 nm (Seinfeld and Pandis, 2006). The refractive index of OM (as BrC) is not known so we considered three scenarios: strongly absorbing (1.65-0.03i at 550 nm), moderately absorbing (1.65-0.003i at 550 nm), and non-absorbing OM (1.65 at 550 nm) (Feng et al., 2013). Although the refractive index has dependence on the wavelength between 350-870 nm, we tried to select the 550 nm as a case to test how OM coating influence sulfate particles in Arctic air.

BHCOAT Mie code by Bohren and Huffman (1983) was used to calculate the optical properties, including scattering cross section (SCS), absorption cross section (ACS), and single scattering albedo (SSA), assuming a core-shell structure. We firstly calculated these parameters assuming a sulfate core and OM shell structure only (ignoring some of the particles that contain soot core). Because the Mie code only can calculate the core-shell structure or homogeneous models, we assume sulfate as a core and OM as a shell in individual particle to build the core-shell model. Based on the core-shell standard mode (Li et al., 2016), we can calculate optical properties of individual internally mixed particles.

**2.6 Back trajectories of air masses and Lagrangian particle dispersion model**

Three-day (72 h) back trajectories of air masses were generated using a Hybrid Single Particle Lagrangian Integrated Trajectory (HYSPLIT) model at the Chinese Arctic Yellow River Station during August 2012, at an altitude of 500m above sea level (Figure 1). Most air masses originate in the Arctic Ocean, and are restricted to this vast marine region during the sampling periods. Based on the TEM observations, air masses from North America and Greenland brought abundant sulfate particles into the sampling area in summertime.

In order to determine the particle origins, the lagrangian particle dispersion model FLEXPART-WRF 3.1 (Brioude et al., 2013) was used. The FLEXPART-WRF model is using meteorological parameters from WRF dynamical simulation. The domain resolution is 50×50 km with 64 vertical levels. The FLEXPART-WRF simulations were launched in a backward mode over 10 days, with the Chinese Arctic Yellow

River Station as an origin. For each simulation (one per sample), 20000 pseudo-particles were released in a small volume around the station position. Each single particle position evolution backward in time was determined by Lagrangian dispersion calculation. Based on the TEM experiments and back trajectory of air masses (Figure 1), we found that there were more S-rich with OM coating particles in the samples collected on August 11, 12, 14 and 15, 2012. Therefore, we further did the FLEXPART-WRF simulation of these four days (Figure 2). The emission intensity in the Arctic area has been also shown in Figure S2.

## 3. Results

### 3.1 Composition and sources of aerosol particles

We summarized average elemental weight and frequency of individual Arctic particles derived from the TEM/EDX. The result shows that O, Na, S, and Cl in individual particles are dominant elements (Figure S3). On basis of the composition and morphology of individual particles, we classified the particles into four major groups: Na-rich (i.e. NaCl, $Na_2SO_4$, and $NaNO_3$), S-rich (i.e. ammonium sulfate and sulfuric acid), and carbonaceous (soot and OM). The classification criteria of different particle types and their sources have been described in a separate study (Li et al., 2016). S-rich particles representing secondary inorganic particles (e.g., $SO_4^{2-}$, $NO_3^{-}$, and $NH_4^{+}$) are transformed from gaseous $SO_2$, $NO_x$, and $NH_3$. OM can be divided into primary organic matter (POM) and secondary organic matter (SOM). SOM is produced from the chemical oxidation of volatile organic compounds (VOCs) and often exhibits OM coating on S-rich particles. Na-rich particles in the marine air are from sea spray and have typical near cubic shape. Soot particles, which contain C with minor O, appear as a chain-like aggregate of carbon-bearing spheres. Our previous study well characterized aging mechanism of sea salt particles in summer Arctic air (Chi et al., 2015). Here we focused on S-rich, soot, and OM particles as the major non-sea salt particle (NSS-particle, 39±5%) in the analyzed samples, which are approximately 29±7% of 2002 particles (Figure 3).

### 3.2 OM coating on sulfate particles

TEM observations revealed a common core-shell mixing structure in fine sulfate
particles (Figure 4a). Elemental mapping of such internally mixed sulfate particles
shows C signals in the coating (C map, Figure 4b) and S and O signals in the center (S
and O map, Figure 4c, d). The elemental line profile of a sulfate particle also shows
sulfate core and C coating (Figure S4). Furthermore, ion maps of individual particles
from the NanoSIMS further exhibit $^{12}C^{14}N^-$ signals in the coating (red color in Figure
4e, f) and $^{32}S^-$ signals in the core (green color in Figure 4e, g). These results provide
strong evidence that the coating is OM and the core is sulfate.
A majority of 781 analyzed NSS-particles (74% by particle number) have a sulfate
core and OM coating (Figures 4 and 5). ~20% of them also contain small soot
inclusions but they only appeared in organic coating, rather than as the core mixed in
sulfate (Figure 5b). The mixing structure is different from our previous findings in
polluted air that soot is normally mixed with sulfate instead of OM coating (Li et al.,
2016). Moreover, we noticed that a few chain-like soot aggregates (1.3% in all
analyzed particles) (Figure S5) only occurred in three samples during the sampling
period (Table S1). Considering the remoteness of the sampling site, such fresh soot
particles are likely to be of local origin, including shipping and flaring (Gilgen et al.,
2018; Peters et al., 2011). Indeed, we found a few of ships moving in Arctic Ocean
during these days from the Ny-Ålesund town.
TEM observations showed that some sulfate particles had unique morphology that
a sulfate particle was surrounded by some smaller particles (Figure 5a). They are
often called "satellite" particles as they were distributed from the central particles
when impacted on the substrate during sample collection. 16% of the analyzed sulfate
particles with satellite particles as shown in Figure 5a were detected in the samples
(Table S1) collected during 9-15 August. NanoSIMS analysis further provided more
information that the satellite particles selected from the samples (Table S1) have
strong $^{32}S^-$ (Figure 6a, c) and $^{16}O^-$ signals (Figure 6d) as well as weak $^{12}C^{14}N^-$ signals
(Figure 6a, b). The CN$^-$ signal normally can represent organic aerosols (Chi et al.,
2015; Ghosal et al., 2014). Previous studies showed that the similar satellite particles
are normally considered as acidic sulfate (Buseck and Posfai, 1999; Iwasaka et al.,
1983). Therefore, we can conclude that these acidic satellites not only contain sulfuric
acid but also some OM or organic acids. Indeed, Fu et al. (2008) found that polyacids
are the most abundant organic compounds, followed by phthalates, aromatic acids,
and fatty acids in Arctic aerosol particles. As a result, these Arctic sulfate particles
with satellites contain certain amounts of sulfuric or organic acids with liquid phase.
Back trajectories of air masses and FLEXPART both shows abundant sulfate particles
and some containing satellite particles were transported from Greenland and North
American (Figures 1 and 2).
AFM was used to obtain 3D image of individual secondary particles impacting on
the substrate. Figure 7a shows that the secondary particles normally have smooth
surface which is different from uneven surface of the Arctic fresh and aged NaCl
particles (Chi et al., 2015). Furthermore, we observed particle thickness through
tilting the specimen stage up to $75^{o}$ in SEM. Figure 7a-b both shows that the
secondary particles look like thin pancake sticking on the substrate. Furthermore, the
sections of two secondary particles in the AFM images shows that the highest heights
of particles are only 0.15 (green line) and 0.26 (red line) of the corresponding
horizontal size (Figure 7a). Here we can conclude that shape of individual particles
was modified when they impacted on the substrate following the airflow. Therefore,
the measured ECDs of individual particles in TEM images are much larger than the
real particle diameter. To calibrate the particle diameter, we obtained volume of dry
particles on the substrate and then calculated their equivalent sphere diameter (ESD)
in the AFM images (Figure 7c). ESD distribution of the secondary Arctic particles
displayed a peak at 340 nm, ranging from 100 nm to 2000 nm (Figure 7d). The core
particles, as sulfate or soot, had a peak at 240 nm and 120 nm, respectively (Figure
7d). In the core-shell particles, we knew size in all the analyzed particles and further
calculated volume of sulfate, OM, and/or soot within individual particles. We can
estimate that OM on average accounted for 63±23% of the dry sulfate particle volume.
Our result shows that the OM volume increases following the particle size increase
(Figure S6).

**4. Discussion**

4.1 **Mixing mechanism of organic, soot, and sulfate**

Lagrangian particle dispersion modeling using the FLEXPART-WRF 3.1 showed that air masses arriving at the sampling site during our field measurement periods were likely originated from the Greenland and North America (Figure 2). Previous studies reported that air masses from North America or Greenland during the summer contain higher concentration of black carbon, OM, and sulfate (Burkart et al., 2017; Chang et al., 2011; Fu et al., 2008; Moore et al., 2011; Park et al., 2013). Indeed, there is strong emission intensity of OC and $SO_2$ around the Arctic area from emission simulation as shown in Figure S2. However, Weinbruch et al. (2012) observed soot particles when cruise ships were present in the area around Ny-Ålesund town. It is possible that minor soot particles are sourced from the ship emissions and most of them are transported from out of Arctic area in the free troposphere (Figure S2).

The sulfate core-OM shell structure observed in the Arctic summer atmosphere is similar to those in the background or rural air in other places (Li et al., 2016; Moffet et al., 2013). Based on the images from electron microscopies, we can infer that OM coating thickness in the Arctic air was comparable with them in rural places but higher than them in urban places. During the transports, organic coatings on sulfates were considered as the secondary organic aerosols and their masses increase following particle aging and growth (Li et al., 2016; Moffet et al., 2013; Sierau et al., 2014). Figures 1 and 2 show that most of particles in the air masses transported long distance from North American. The result indicates that these long-range transportation of secondary sulfate particles have enough time to experience the possible atmospheric heterogeneous reactions on particle surfaces or cloud processes in the Arctic air. Similarly, Moffet et al. (2013) found that soot inclusions occurred in OM coating when OM coating on sulfates built up through photochemical activity and pollution buildup the Sacramento urban plume aged. On the other hand, the sulfate/OM particles with soot inclusions are probably formed in a similar way as those found elsewhere (Li et al., 2016) – e.g., soot particles may have acted as nuclei for secondary sulfate or organic uptake during their transports (Riemer et al., 2009).

Similarly, besides the OM coating in the Arctic particles, Leck and Svensson (2015)
found some biogenic aerosols like gel-aggregate containing bacterium in ultrafine
particles. However, we didn't find any gel-like particles in the samples because our
sampler had very low efficiency for ultrafine particles.
TEM images show that most of the internally mixed sulfate particles display
sulfate core and OM coating on the substrate (Figures 4a and 5b, c). The sulfate and
OM separation in individual particles were defined by You et al. (2012) as
liquid-liquid phase separation (LLPS). Concerning the knowledges of the LLPS can
better understand particle hygroscopicity, heterogeneous reactions of reactive gases on
particle surface, and organic aging (You et al., 2012). They also reported that the
LLPS can reflect the O:C ratio in the OM, which is roughly $\leq 0.5$. In this study, we
did observe the LLPS in almost all the fine sulfate particles, which indicates that the
secondary OM in the coating might be not highly aged. Therefore, we speculate that
the thick OM coatings were consistently built up during the long-range transport of
sulfate particles and part of secondary OM in the coating likely formed in Arctic area.
Indeed, some studies reported that there are various sources of organic precursors
during the Arctic area, such as biogenic VOCs from ice melting and open water
(Dall Osto et al., 2017) and anthropogenic VOCs from shipping emissions in
summertime (Gilgen et al., 2018). The dependence of OM volume on particle size
(Figure S6) suggests that the suspended sulfate particles are initially important surface
for secondary OM formation. Moreover, the common OM coating on sulfate particles
indicates that secondary OM as the surfaces of fine particles might govern the
possible heterogeneous reactions between reactive gases and sulfate particles in the
Arctic air.
It should be noted that most of secondary OM not only occurred on the surfaces
of sulfate particles but also its mass (mean mass at $63\pm23\%$) dominated in individual
particles (Figure 7d). The OM dominating in individual particles can influence the IN
and CCN activities of secondary sulfate particles (Lathem et al., 2013; Martin et al.,
2011). For example, some studies found that an increase in organic mass fraction in
particles of a certain size would lead to a suppression of the Arctic CCN activity
(Leck and Svensson, 2015; Martin et al., 2011). Moreover, OM as particle surfaces
can significantly influence hygroscopicity and IN activity of sulfate particles (Wang et
al., 2012).

**4.2 Potential impact of OM on optical properties of sulfate-containing particles**
The internal mixing of soot, sulfate, and OM can change optical properties of
individual particles in the atmosphere. Recent studies showed that BrC has been
detected in the OM in the polluted and clean air and even in upper troposphere
(Laskin et al., 2015; Wang et al., 2018). Feng et al. (2013) further calculated the
contribution up to 19% of the optical absorption of the strongly absorbing BrC in
global simulations which is after the absorption BC aerosols. Although we didn't
directly measure the optical absorption and BrC in the Arctic atmosphere, various
colored OM (e.g. nitrated/polycyclic aromatics and phenols), referred as BrC, were
detected in the Arctic atmosphere in different seasons (Fu et al., 2008;
Wöhrnschimmel et al., 2013; Zangrando et al., 2013) and in surface ice or snowpack
(Browse et al., 2013; Doherty et al., 2013; Hegg et al., 2010). We also noticed that the
$^{12}C^{14}N^-$ signal generally occurred in all analyzed OM coating in sulfate particles
(Figure 4e-f). Herrmann et al. (2007) considered that $^{12}C^{14}N^-$ from NanoSIMS
represents nitrogen-containing organic in the detected materials. In this study,
although we could not determine that all the organic materials in the OM coating were
nitrogen-containing OM, the NanoSIMS data as shown in Figure 4 indicated that the
OM coating more or less homogenously contained nitrogen-containing OM. As a
result, the nitrogen-containing OM indicates that the OM coating could contain
certain amounts of secondary BrC (Jiang et al., 2019; Laskin et al., 2015).
To understand how OM coating influence optical properties of sulfate particles,
we assume three scenarios of OM coating as BrC: strongly absorbing (case 1),
moderately absorbing (case 2) or non-absorbing OM (case 3) with a refractive index
of 1.65-0.03i, 1.65-0.003i, and 1.65 at 550 nm according to *Feng et al.* (2013). Based
on the size measurements shown in Figure 7d, we can calculate volume of sulfate and
OM within each particle. We input volume of each component and the corresponding
refractive index into the Mie code and then calculated optical properties of individual
sulfate particles in the samples. Based on optical data statistic of 575 particles, Figure
8a show that the OM coating is strongly absorbing BrC (referred to case Abs1), as by
Feng et al.(2013), the average absorption cross section (ACS) of individual particles is
estimated to be $2.67 \times 10^{-14}$ $m^2$. This value is 8.30 times higher than the aerosol ACS
($3.22 \times 10^{-15}$ $m^2$) when assuming that the BrC is moderately absorbing (referred to case
Abs2, Figure 8a). However, the scattering cross section (SCS) of individual particles
only shows a small change (Figure 8b). Figure 8c also shows that the single scattering
albedos (SSAs) of individual particles are 0.92, 0.99, and 1 when assuming the BrC as
strongly, moderately and non-absorbing (cases SSA1 to SSA3). These results suggest
whether we consider organic coating as BrC may have a significant influence on the
absorption properties of individual sulfate particles.
In this study, we expored the relationship between ACS of individual particles and
particle diameters. Interestingly, Figure 8d shows that ACS of individual fine
OM-coating sulfate particles increased following the increasing particle size. The
result shows that the ACS can be enhanced following particle size growing and
particle aging. In other word, OM-coating sulfate particles transported more longer
distances and they might have stronger optical absorption in the Arctic air.
Current climate models estimated the radiative force of Arctic BC (Sand et al.,
2013; Shindell, 2007; Winiger et al., 2017; Zanatta et al., 2018), but none specifically
considered optical properties of Arctic BrC. Our study well revealed OM coating on
sulfate particles and this detail microphysical complexity of aerosol particles will be
useful to construct the atmospheric radiation and CCN/IN simulation in Arctic
atmospheric models in the future.

**5 Summary**
Different individual particle techniques, such as TEM/EDS, STEM, SEM,
NanoSIMS, and AFM, were applied to study S-rich, soot, and OM particles in the
Arctic air in summer. Sulfate particles accounted for approximately 29±7% by
number of all analyzed particles in Arctic air. TEM and NanoSIMS commonly

observed OM coating and sulfate core individual sulfate particles, defined as the LLSP. The common OM coating on sulfate particles indicates that secondary OM as the surfaces of fine particles might govern the possible heterogeneous reactions between reactive gases and sulfate particles in the Arctic air. Moreover, 20% of them also contain small soot inclusions but they only appeared in organic coating, rather than as the core mixed in sulfate. The mixing structure is totally different from the previous findings that soot is internally mixed with sulfate instead of OM coating in urban polluted air.

Size distribution of the secondary Arctic particles displayed a peak at 340 nm, ranging from 100 nm to 2000 nm. The core particles, as sulfate or soot, had a peak at 240 nm and 120 nm, respectively. Furthermore, we can estimate that OM on average accounted for 63±23% of the dry NSS-particle volume. Based on microscopic measurements of individual particles, we not only built up one core-shell model but also quantify volume of OM and sulfate in individual particles. The Mie code was used to calculate optical properties of internally mixed sulfate/OM particles when we considered OM as non-absorbing, moderately absorbing BrC, and strongly absorbing BrC. We found that the aerosol ACS is 8.30 times higher than the BrC as moderately absorbing. We concluded that whether we consider organic coating as BrC may have a significant influence on the absorption properties of individual particles in the Arctic air. Moreover, individual fine OM-coating sulfate particles increased following the increasing particle size. Therefore, we proposed that further studies should focus on the BrC in Arctic aerosols: What mass concentrations of BrC are in fine particles? What kinds of BrC are in fine particles? The optical mass absorption of BrC in fine particles should be investigated? These results can be used to evaluate how BrC aerosols influence the Arctic climate.

**Author Contributions:** WL and ZS designed the study. YZ and XS collected aerosol
particles. WL, HY, and JZ contributed laboratory experiments and data analysis. HY
and WL performed optical calculation and wrote part of first draft. PT and MD
provided the online measurement data of new particle formation and growth. JS and
XZ coordinated the field campaign. All authors commented and edited the paper.

**Competing interests:** The authors declare no competing financial interests

**Acknowledgments** We thank Boris Quennehen to provide data from the
FLEXPART-WRF. This work was funded by National Natural Science Foundation of
China (41622504, 41575116, 31700475) and the Hundred Talents Program in
Zhejiang University, Z.S. acknowledges funding from NERC (NE/S00579X/1).

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

**Figure Captions**

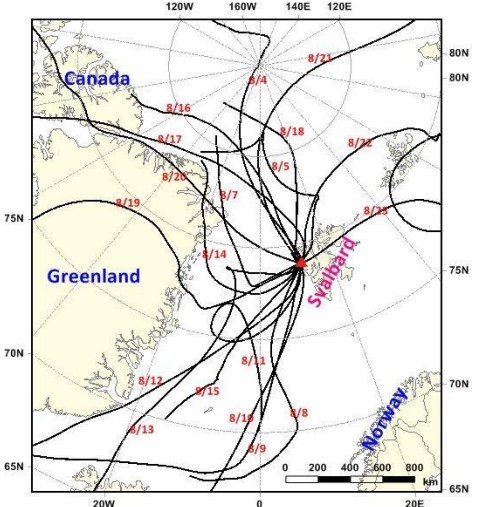

**Figure 1** 72 h back trajectories of air masses at 500m over Arctic Yellow River Station in Svalbard during 3–26 August 2012, and arriving time was set according to the sampling time

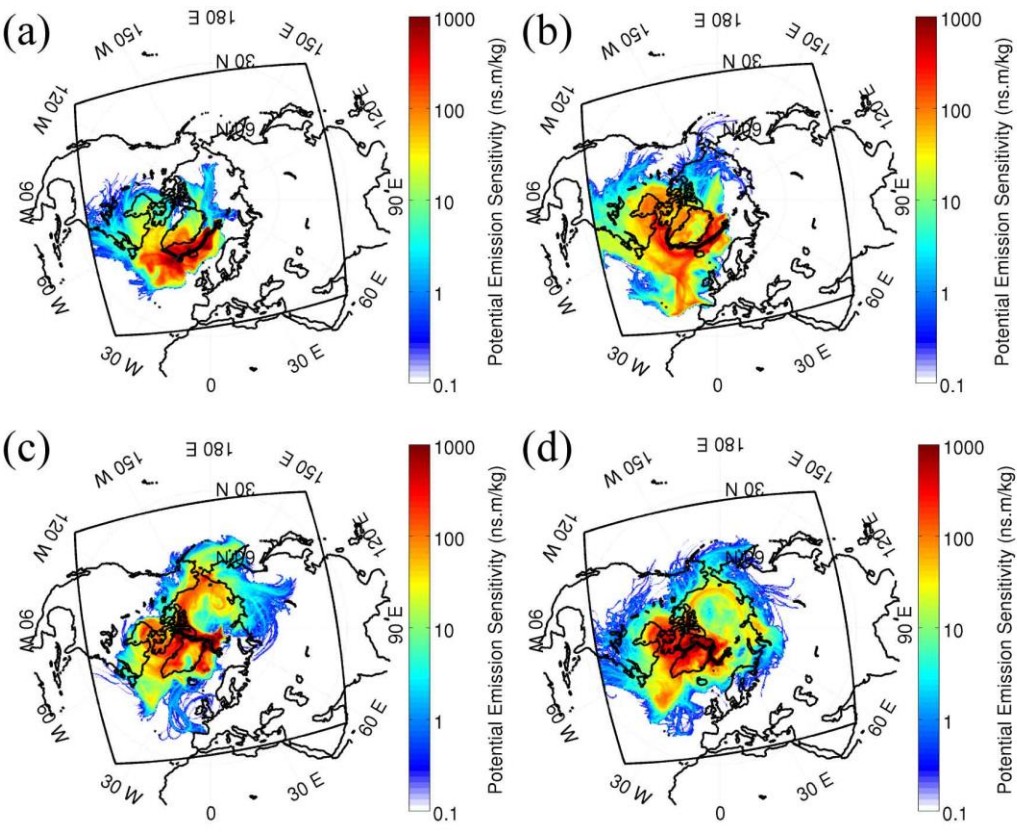

**Figure 2** FLEXPART-WRF PES on August 11, 12, 14, and 15, 2012. Black square is showing the WRF domain used to initiate the FLEXPART-WRF simulation.

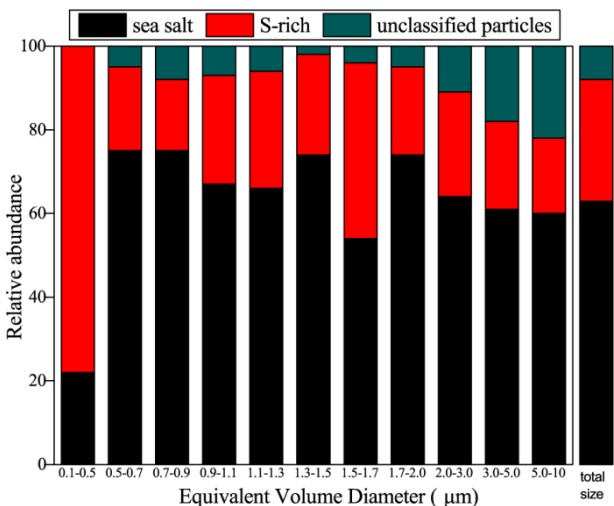

**Figure 3** Morphology and relative abundances of typical individual aerosol particles in the 21 analyzed samples.

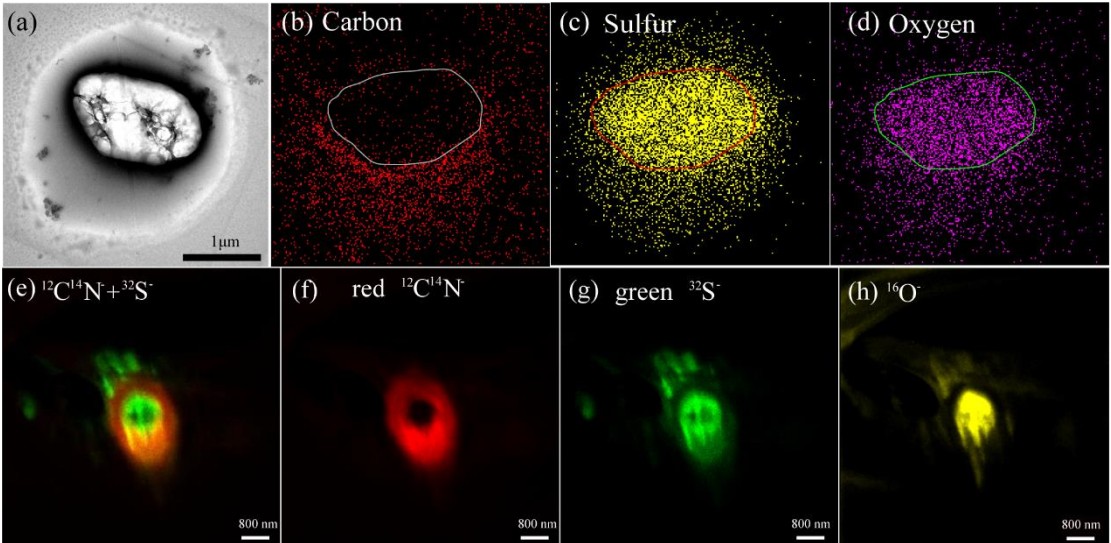

**Figure 4 TEM Observations of a secondary particle and NanoSIMS intensity threshold maps of an aerosol particle with sulfate core and OM coating.** (a) Bright-field TEM image of an internally mixed particle; (b) elemental carbon (c) sulfur and (d) oxygen maps of the internally mixed particle shown in 1(a); (e) Overlay of $^{12}C^{14}N^{-}$ and $^{32}S^{-}$ ion maps in an internally mixed particle; (f) CN$^{-}$ map (g) S$^{-}$ (h) O$^{-}$ secondary ion maps. Ion maps with a set of aerosol particles were shown in Figure S1.

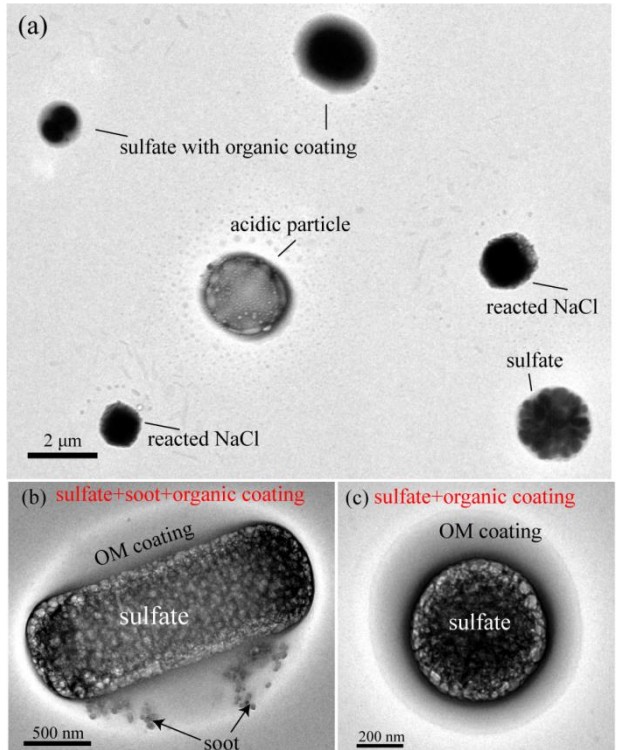

**Figure 5 TEM images of individual particles containing sulfate, OM, and soot.** (a) Low magnification TEM image showing sulfates, sulfate with OM coating, and reacted NaCl particles. (b) an internally mixed particle of sulfate and soot with OM coating (c) a particle with sulfate core and OM coating.

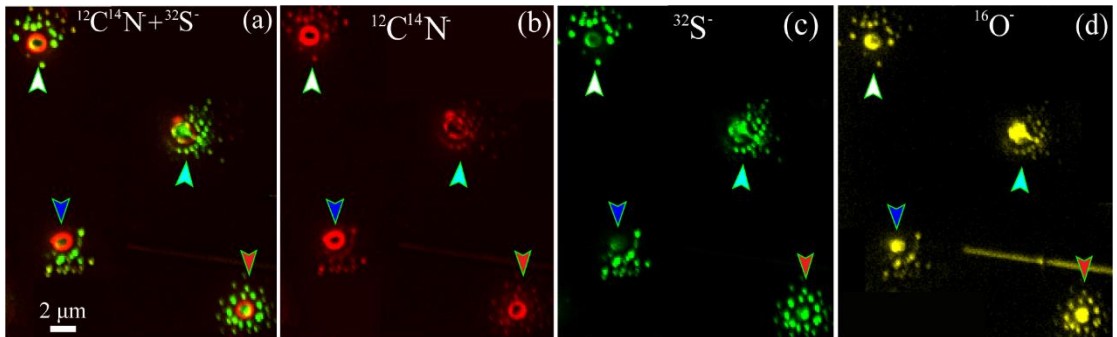

**Figure 6 NanoSIMS intensity threshold maps of individual aerosol particles surrounded by satellite particles.** (e) Overlay of $^{12}C^{14}N^-$ and $^{32}S^-$ ion maps of individual particles. (f) CN$^-$ (g) S$^-$ (h) O$^-$ maps. Four particles were indicated by white, pink, blue, and red arrows.

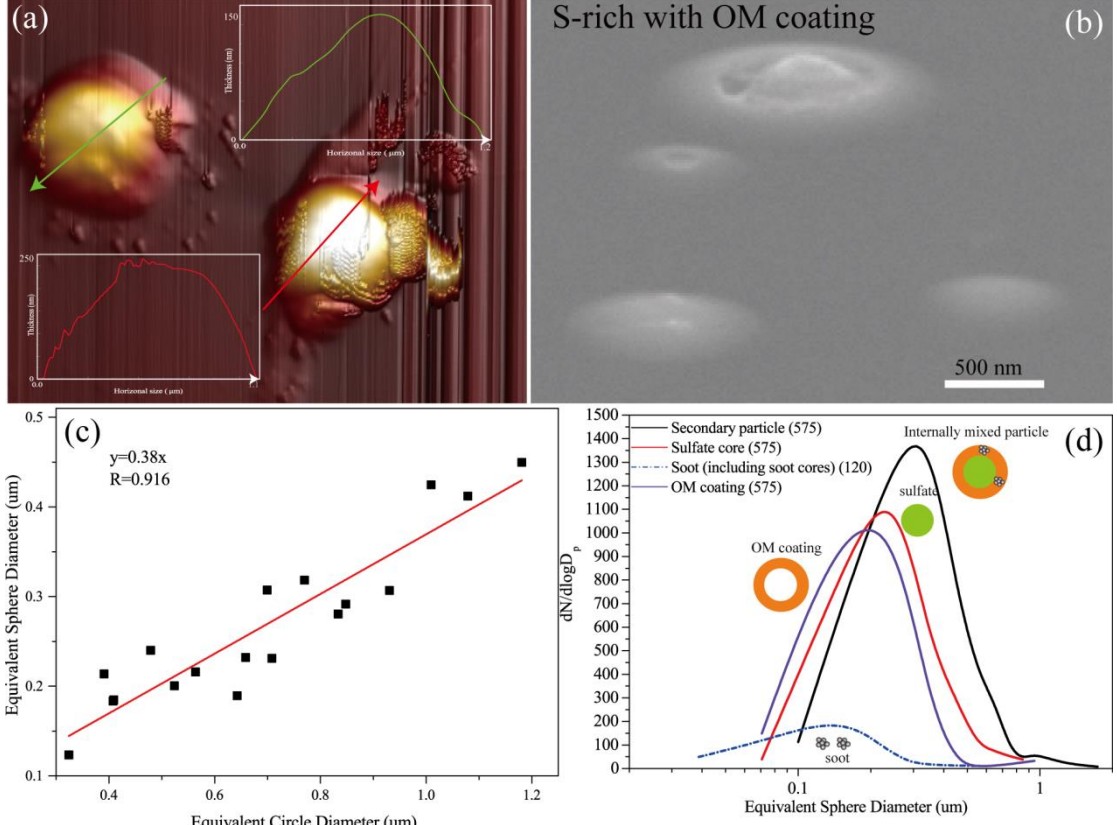

**Figure 7** Secondary particles on the substrate. (a) 3-D AFM image of secondary sulfate particles. The colorful

arrows represent particles surface properites of the particle section. (b) SEM image of S-rich with OM coating

obtained from 75° tilt of the SEM specimen stage (c) The near linear relationships between ECD and ESD based

on S-rich particles with thick OM coating by Atomic force microscopy. (d) Size distribution of individual particle

with OM coating and sulfate cores based on the estimated ESD diameter from TEM image. Sizes of soot particles

are equal to the equivalent circle diameter.

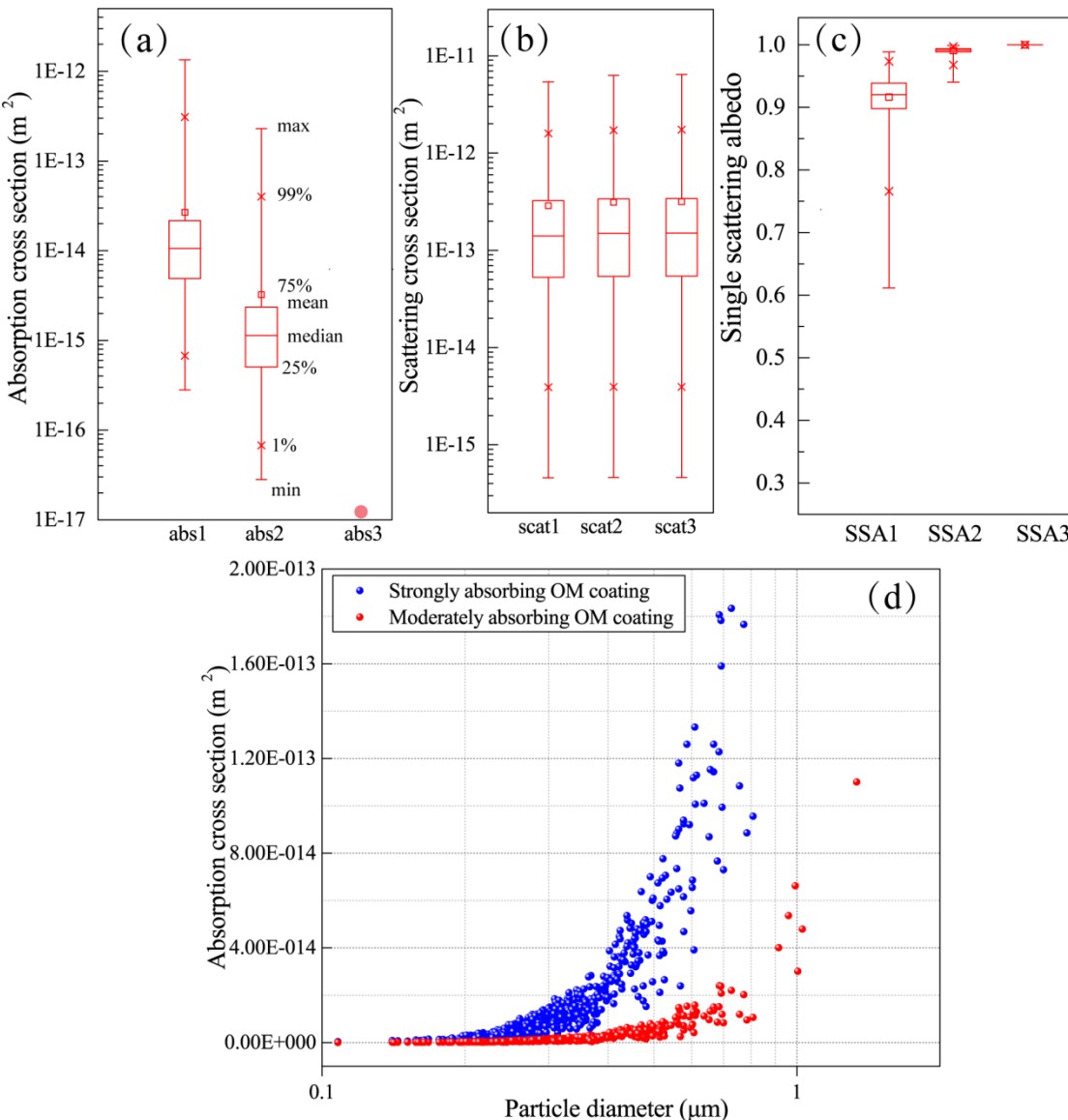

**Figure 8 Optical properties of Box-and-whisker plots showing optical parameters of all analysed particles assuming sulfate core and BrC shell (not considering soot cores in the particles)**. (a) Scattering cross section (b) Absorption cross section (c) Single scattering albedo. Top to bottom makers in the box-and-whisker represent max, 99%, 75%, mean, median, 25%, 1%, min values. (d) Absorption cross section along with particle diameter assuming strongly absorbing BrC and Moderate absorbing BrC as the particle OM coating.