# Peer review of "Organic coating on sulfate and soot particles during late summer in the Svalbard Archipelago"

_Atmospheric Chemistry and Physics, 2018_

## Referee Comment (RC1) · Anonymous Referee #1 · 27 Feb 2019

Yu et al. present findings from detailed compositional measurements of Arctic aerosol in Svalbard during August 2012. While there is obvious importance of conducting detailed physiochemical characterizations of Arctic aerosol in terms of their radiative impacts and subsequent indirect effects on frozen surfaces, there are major issues with the manuscript by Yu et al. that would need to be addressed prior to publication. These issues stem from possible misinterpretation of the data that shape the reported main findings. It would behoove the authors to provide a sufficient level of detail on the methodologies (including caveats) and results to support the main conclusions they report.

**General comments:**

There is a scarcity of detail regarding which samples and particles were analyzed. More specifically, which samples were analyzed, which particles were analyzed per sample and how those were chosen, how many particles per sample were analyzed, and why only select samples and particle numbers were analyzed under each method is not at all defined. For example, which of the samples constituted the 2002 and 575 particles analyzed by TEM and TEM/EDS, respectively? For certain techniques, only a few samples (i.e., 3 samples for NanoSIMS but no mention of particle number) or even only a handful of particles (i.e., only 17 particles for AFM but no mention of which sample(s) these came from) were analyzed, and in the case of SEM there is no information on sample or particle number. I understand that some of these tools, i.e., AFM, are time-consuming which is why a low number of particles were analyzed, but then the authors need to be careful about overstating result interpretations. It is important to know how many particles and from which samples to provide sufficient statistics and afford information on daily source variability. As it stands, there is no way to tell how representative the percentages (which are hidden in the text) are of summertime aerosol in general, or just of specific samples from select days.

It would be helpful to provide a figure or two of the overall picture of aerosol composition, e.g., bar graphs or pie charts. There are percentages provided in the text, but showing the relative abundance of each particle type is pretty standard. Along relative abundance, the authors report that 29% of the particles were non-sea salt. What percentage were unclassified? What percentage is the "majority of NSS-particles"? As a result, it is not clear how important these particles are in general in the context of radiative impacts, given sea salt was what seemed like the dominant particle type and also largely affects scattering and SSA. In addition, to demonstrate that these particle types are important for the Arctic energy budget, they should show extinction properties for total aerosol (including sea salt) in Figure 6. I would think given the typical sizes for these types of particles (sulfate and soot) and reported abundance for this particular study, they would not affect the scattering cross sections relative to sea salt. The emphasis on the radiative impacts of these aerosol types is a large part of the manuscript, so their properties need to be presented in broader context. Are they important within the total aerosol population or not? This would be more relevant to the actual atmospheric implications.

There are several issues with the methods as presented. For instance, there is very little detail given on how the particles were classified under each technique, there are no errors or statistical analyses reported, and certain methods have very little specification detail (i.e., SEM is a very short paragraph). Regarding the samples storage, at 20% RH, I would assume all volatile and most semi-volatile species would evaporate, significantly altering particle shape, size, and composition. I will admit, these techniques are not my area of expertise, but the authors should at least comment on potential losses and caveats with this storage method. If there are significant losses of material, how representative are the analyzed particles of the total ambient aerosol population at the time of collection? I am skeptical the authors are comparing apples to apples by possible alteration of particles during storage. Regarding the source analysis, there is very little detail given on the FLEXPART modeling and only a couple sentences on the results and discussion of the simulations. It is used to a very minimal extent and very generally summarized, even though Arctic aerosol

sources can vary drastically day-by-day, and especially given possible local contributions. Figure 5 is very difficult to discern and glean any source information from it. Also, why are only these particular days shown?

There is no background on previous relevant studies conducted at the study location, even though there is a long-term monitoring station with aerosol measurements at Ny-Ålesund (https://www.esrl.noaa.gov/psd/iasoa/stations/nyalesund). It is not the same exact location as the Chinese site, but close enough to at least use those routine, publically-available measurements to provide some broader spatial and temporal context.

There is only basic mentioning of biogenic VOCs, but none of biogenic or biological aerosol. The Arctic summer, especially in remote coastal sites, is largely affected by gases and aerosol from primary productivity due to the availability of sunlight and open water. There is no discussion on if the OM is biogenic/biological, and in general, the definition of OM is vague.

**Technical corrections:**

There are many typos, grammatical issues, and a lack of necessary explanation (e.g., the 3 sets of bars in Figure 6, SEM a short paragraph with no numbers, etc.).

---

## Referee Comment (RC2) · Anonymous Referee #2 · 3 Mar 2019

**Overall Comments:** The authors use multiple measures to investigate the physical properties of non-sea salt aerosols collected in the Arctic. They qualitatively describe the mixing state of organic, soot, and sulfate aerosols. Although the core data set of this manuscript is informative, as is, the presentation and discussion are confusing and possibly misleading. No large scale descriptive statistics are provided, the methods and presentation are often unclear and/or redundant, the environmental context is lacking, and the optical conclusions are not well explained. This manuscript would benefit from additional analysis, clarification of methods, adding proper context to results, and careful grammar review.

**General Comments:**

*Unclear Sample Selection*: In multiple sections of this paper the authors discuss a subset of their samples without explaining why they were chosen. This detracts from their discussion and leaves the reader questioning what is not being discussed. Anytime a subset of samples is chosen or used, an explanation needs to be provided explaining why these are the best samples for that specific analysis or investigation. A few example occurrences can be found in lines 275, 281, 118, 190.

*Environmental Context:* The only environmental context explored in this text is the back trajectories provided for 4 of the sampling days. Whenever field data is discussed, external variables such as temperature, humidity, and back trajectories can help explain variability observed in the data. To improve this work, the context of the sampling should be explored as a driving factor between differences in particle characteristics.

2.1 I am also not convinced that the daily averaged back trajectory calculations are meaningful for these samples. (I'm also not sure the trajectories were daily averaged as the methods section just said they were calculated for a given day, not the frequency of times or given time on the day.)   Because the sample length was between 20 minutes and 2 hours, a back trajectory occurring during the actual sampling time for all samples would be more appropriate.

*Optical Property Calculations:* The authors attempt to estimate the potential radiative forcing implications of their results using a core-shell model and Mie theory.   Although this is a potentially meaningful result, the methodology needs to be discussed further and improved.

3.1 When performing Mie theory calculations the input size distribution characteristics and wavelength dependence (angstrom exponent) have dramatic effects on the results. These data need to be reported. The authors state that a single size distribution is used.  How was this size distribution established?  Could you run your analysis over all of the size distributions you observed to estimate the variability?

3.2 The way the authors include soot in their calculations is not appropriate, as soot is treated as a core with a mixed sulfate-OM shell.  This is in direct conflict with their observations that soot is only observed to be associated with the OM shell and sulfate is always the core. It would be better not to include a soot calculation than include a misleading one.

3.3 The relevancy of the refractive index (RI) choice needs to be discussed further. The authors use an RI from biomass burning brown carbon as their slightly absorbing case but frequently state that they believe the OM to be secondary. If that is the case, their estimations are likely too high for all cases but the non-absorbing case. The authors should include refractive indexes from secondary organic aerosol brown carbon to get a more realistic answer. These are generally much lower than the RI values that they used. Additionally, the scattering component chosen (1.65) is relatively high and not explained. Please explain why this value was chosen as it is especially important in core-shell cases.

3.4 Stemming from the above comment, the relevancy of each refractive index case needs to be discussed in the context of the Arctic. Which case seems the most likely?

3.5 The authors use m=1.55 for sulfate. RI values have wavelength dependence, was this included in any way?

3.6 The authors average the absorption cross section on a per particle basis. This isn't meaningful since observed absorption cross sections will depend on the whole aerosol population (and the size of the particle). The authors should calculate an absorption cross-section for the ambient aerosol concentration during their sampling and compare it to other absorption observations.

***Redundancy and Clarity:*** The writing of this paper needs improvement. There are multiple times when the authors restate the same point twice or fail to introduce a topic before discussing their results. An example of redundancy can be observed when Copper TEM grids are introduced in lines 120, 140, and 156. An example of an improper discussion occurs in the discussion of satellite particles, which are introduced in line 280 with no context or explanation. This forces the reader to infer what the authors mean by satellite particles, possibly leading them to wrong conclusions. To improve the manuscript, I recommend careful reconsideration of the presentation of the data, with special consideration to avoiding redundancy and ensuring the appropriate context is present.

4.1 There are multiple points in the methods, a few of which I've included in the specific comments, that are unclear or confusing. The reader needs to be able to understand exactly which analysis was performed on which filters for how many particles if they are to believe your result.

***Sample Information***: The only summary of the total data set is provided in figure S6 and this figure states that only 3 samples (of 46 collected and 21 analyzed). No explanation is given as to why is summary is so limited given the authors have EDS data (which they used for classification) of 20-30 particles over 21 samples (at least 400 data points). All the of following discussions only make sense if they are provided in the context of overall sample composition.

**Specific Comments:**

There are many grammatical errors and redundancies that I have not addressed below.

The phrase internal mixing is used throughout the manuscript without an explicit definition. An explanation of what exactly you mean when you say something is "internally mixed" would improve the manuscript

**Line 55:** Change "Artic" to "the Artic"

**Line 60:** Change "nature" to "natural"

**Line 64-66**: Restructure this sentence for clarity. Treat the percentages in a consistent way as to not confuse the reader. For example, this sentence could be changed to:" For example, Winger et al.(2017) showed most Arctic BC is sourced from domestic activities (35%) and transportation (38%), with only minor contributions from gas flaring (6%), power plants (9%), and open fires (12%)"

**Line 72:** I'm not sure exactly how this sentence fits in with the brown carbon theme of this paragraph. Are these compounds commonly found in brown carbon or organic aerosols in general? Please add some context.

**Line 84:** replace "were" with "have been"

**Line 96-68:** This sentence is confusing. Please rewrite it more concisely and clearly.

**Line 99-100:** Change "collected on 7 to 23 August, 2012 in the Arctic." to "collected in the Artic between August 7th and 23rd, 2012."

**Line 104:** replace "on substrate" with "on a substrate"

**Line 118:** Change "samples between 7 and 23 August, 2012." to "samples collected between August 7th and 23rd, 2012."

**Line 119:** Replace "analyzed for TEM analysis" with "analyzed with TEM"

**114-131:** Restructure your sampling section. As written the reader may think that you sampled with 2 separate samplers an individual particle sampler and cascade impactor. After reading the paper, there is only one sampler. This confusion can be remedied by introducing the cascade impactor earlier.

**122-123:** Add the top size cutoff for this sampler.

**138:** replace "within a" with "for a"

**148:** 2002 particles examined over all the samples, or in a specific filter?

**151:** Clarify this sentence. Do you mean to say you ", we only checked elemental compositions of 20-30 particles" in each sample?

**155-156:** The statement about Cu is redundant. This has already been stated in line 120 and 140.

**156-157:** What is the difference between what is stated here and what is stated in lines 147-149?

**162-163:** Replace "is **the** image analysis platform…" with "is **an** image analysis platform".

**177:** Replace "Organic Matters" with "Organic Matter"

**Line** 184: Replace "TEM grids was" with either "The TEM grid was" or "TEM grids were"

**Line 192-194:** Incomplete sentence

**Line 194:** I stopped making basic grammar and structure critiques at this point.

**Line 222**: Does treating this as a core-shell system with BC in the middle and sulfate and OM mixed on the outside have any basis? You've indicated that you have soot inclusions on the outside of predominantly sulfate particles, so why would soot be on the inside?

**Line 224:** This sentence reads as if you've calculated the refractive index of the particles. Did you measure the optical properties of these particles?

**Line228:** "In this study". It is unclear whether this is referring to your previous work or this manuscript.

**Line 253:** Is it possible that coagulation of primary organic particles and S-rich particles could have led to the formation of organic coatings? Are you sure assumption that all organic coatings are secondary valid?

**Line 272:** Can you say a percentage of NSS particles that are S-rich with an OM coating? Or a percentage of S-rich particles that have a coating? This would strengthen the paper if an actual number was given.

**Line 274-276**: Are these specific samples special or is there something that you think may have caused the low frequency of soot inclusions? If so, please explain why.

**Line 277:** This statement needs to be better supported. Just because a site is remote does not mean particles are local in origin. If this is supported by your trajectory calculations, mention them here. Also, don't the soot inclusions also imply that perhaps the OM is not secondary in nature? Soot is 100% primary and often co-emitted with primary OM, so if there is soot associated with OM coatings the soot itself is primary and so possibly some of the OM is primary as well.

**Line 280:** You need to define what satellite particles are; you have not discussed or defined them previously. Are they simply splatter of liquid portions of the particle when the particle is collected?

**Line 280-281:** is there a reason why satellite particles would have been observed on these days but not other days?

**Line 281:** This is misleading and implies you performed the NanoSIMS analysis on 11 samples. In the methods section, it says only two samples were analyzed with NanoSIMS.

**Line 287-289:** This is misleading. It reads as if you have done molecular characterization of the organic matter.

**Line 314**: You back trajectories are only for specific days, be transparent about this in the discussion

**Line 317-318:** This conflicts with your earlier comment that most BC should be local.

**Line 326-342**: This discussion reads like a list of facts, but why they are all relevant is not always stated. Explain why each observation is important and how it adds context to your results.

**Line 343:** Why is dry included here? Are there also wet particles that you have not discussed?

**Line 344-347:** Explain LLPS in simpler terms and why it's important.

**Line 346:** It's unclear what 90% is referring to in this statement.

**Line 348:** There's likely a complex relationship between phase state, oxidation state, and humidity. This needs to be investigated and explained further if statements about aerosol age are going to be made. Additionally, shouldn't you see a variety of ages of aerosol in your samples? Showing contrast between aged and unaged particles would be interesting and convincing.

**Line 376-378:** This is circular reasoning because $^{12}C^{14}N$- was what you used to identify OM so of course it was observed in the OM coatings. I don't think this data set is appropriate to make conclusions about the N content of OM coatings. That said, if you were able to calculate the mass concentration of N in the coatings with NanoSIMS that might give you a better indication of the BrC potential of the OM.

**Line 390-393:** The average absorption cross-section is reported on a particle basis. This would be much more meaningful if it was extrapolated to environmental conditions. Because you've sampled from the atmosphere, you should be able to approximate particle concentrations, correct? Is so, you could back calculate this to an actual atmospheric absorption contribution and compare it to expected absorption from other species and measurements. This would be significantly more meaningful.

**Line 413, 260:** 29% number should include a standard deviation.

**Line 410:** The last section shouldn't simply repeat what was stated in the above sections, but instead present the data in additional context and discuss the implications.

---

## Author Comment (AC1) · 10 May 2019

please see the attached file including the responses and revised manuscript.

Please also note the supplement to this comment:
https://www.atmos-chem-phys-discuss.net/acp-2018-1205/acp-2018-1205-AC1-supplement.pdf
* * *

---

## Author Response (AR1)

The file includes

1. Responses to the reviewer#1' comments

2. Responses to the reviewer#2' comments

3. The revised manuscript

**General Response: We thank the reviewer for your helpful comments. We have addressed all comments and provided point by point response below. The revised manuscript is presented in below.**

**Response to the referee#1' comments**

Yu et al. present findings from detailed compositional measurements of Arctic aerosol in Svalbard during August 2012. While there is obvious importance of conducting detailed physiochemical characterizations of Arctic aerosol in terms of their radiative impacts and subsequent indirect effects on frozen surfaces, there are major issues with the manuscript by Yu et al. that would need to be addressed prior to publication. These issues stem from possible misinterpretation of the data that shape the reported main findings. It would behoove the authors to provide a sufficient level of detail on the methodologies (including caveats) and results to support the main conclusions they report.

Response: We carefully addressed all of the questions and concerns raised.

General comments:

There is a scarcity of detail regarding which samples and particles were analyzed. More specifically, which samples were analyzed, which particles were analyzed per sample and how those were chosen, how many particles per sample were analyzed, and why only select samples and particle numbers were analyzed under each method is not at all defined.

Response: We added more details in the Experimental section. In addition, we revised Table S1 and added information on what samples were analyzed by what methods.

which samples were analyzed?

Response: Revised in the context line 144-146

"The sample information such as local sampling date and time and meteorological conditions (e.g., temperature (T), relative humidity (RH), pressure (P), wind direction (WD), and wind speed (WS)) are listed in Table S1."

which particles were analyzed per sample and how those were chosen,

Response: We follow a commonly used methodology in single particle analysis community. Since the distribution of aerosol particles on TEM grids was not uniform, with coarser particles occurring near the center and finer particles on the periphery five areas were chosen from the center and periphery of the sampling spot on each grid to ensure that the analyzed particles are representative of the whole sample.

We added the following in the revised manuscript. Line 159-162

"The distribution of aerosol particles on TEM grids was not uniform, with coarser particles occurring near the center and finer particles on the periphery. Therefore, to ensure that the analyzed particles are representative, five areas were chosen from the center and periphery of the sampling spot on each grid. Through a labor-intensive operation, 2002 aerosol particles with diameter < 10 µm in 21 samples were analyzed by TEM/EDS (Table S1)."

How many particles per sample were analyzed, For example, which of the samples constituted the 2002 and 575 particles analyzed by TEM and TEM/EDS, respectively?

Response: Through a labor-intensive operation, 2002 aerosol particles with diameter < 10 μm in 21 samples were analyzed by TEM/EDS (Table S1). To check composition of individual particles, EDX was manually used to obtain EDS spectra of individual particles. In the clean Arctic air, there are several relatively easy-to-identify particle types including sea salt, sulfate, soot, and OM. Because soot particles have chain-like aggregation, it is not necessary to check their elemental composition. Sea salt particles display spherical or square shapes and are stable under the electron beam in TEM but sulfate particles are spherical but flats on the substrate and produce unstable bubble under the electron beam (Buseck and Posfai, 1999; Chi et al., 2015). TEM observations also can clearly identify sulfate particles or sulfate with OM coating. Therefore, we can identify Arctic particle types based on their morphology. We usually randomly chose 20-30 particles in each sample for elemental analysis to confirm the identification of particle types (Table S1). In total, EDS spectra of 575 particles were manually obtained and saved in the computer for elemental composition analysis. Detailed information is now added to Table S1 We would like to point out that it is not realistic to analyse every single particle collected on the grid as each EDS analysis took about 100 s and all data need to be analysed manually. Therefore, we In the revised manuscript (track changed), we added the following (line 165 to 180)

"In the clean Arctic air, there are simply particle types including sea salt, sulfate, soot, and OM. Because soot particles have chain-like aggregation, it is not necessary to check their elemental composition. Sea salt particles display spherical or square shapes and are stable under the electron beam in TEM but sulfate particles are spherical but flats on the substrate and produce unstable bubble under the electron beam (Buseck and Posfai, 1999; Chi et al., 2015). TEM observations also can clearly identify sulfate particles or sulfate with OM coating. Therefore, we can easily identify Arctic particle types based on their morphology. Because of the time-consuming in the experiment, it is not necessary to frequently check elemental composition of the same particle type. For the data statistic in this study, we randomly checked elemental composition of 20-30 particles in each sample (Table S1). EDS spectra of 575 particles were manually selected and saved in the computer for elemental composition analysis. Particles examined by TEM were dry at the time of observation in the vacuum of the electron microscope. In our study, the effects of water and other semi-volatile organics were not considered as they evaporate in the vacuum."
Information is now added to Table S1 s.

For certain techniques, only a few samples (i.e., 3 samples for NanoSIMS but no mention of particle number) or even only a handful of particles (i.e., only 17 particles for AFM but no mention of which sample(s) these came from) were analyzed, and in the case of SEM there is no information on sample or particle number. I understand that some of these tools, i.e., AFM, are time-consuming which is why a low number of particles were analyzed, but then the authors need to be careful about overstating result interpretations. It is important to know how many particles and from which samples to provide sufficient statistics and afford information on daily source variability. As it stands, there is no way to tell how representative the percentages (which are hidden in the text) are of summertime aerosol in general, or just of specific samples from select days.

Response: Information is given in Table S1.

For some methods, we analysed only a small number of particles. This does not affect our conclusion. The purpose of AFM is to calibrate the equivalent circle diameter to equivalent spherical diameter. As the previous studies (Chi et al., ACP, 2015), the number of samples analysed is enough to address this issue.

The purpose of NanoSIMS is to confirm the OM coating. we analysed 32 S-OM particles, which have the same morphology and composition; analyzing more samples is unlikely adding more information .

Table S1

* number of particles analysed

| Date | Local time | T | RH | P | WD | WS | TEM | EDX | SEM | AFM | NanoSIMS |
|---|---|---|---|---|---|---|---|---|---|---|---|
| 2012.8.7 | 20:50 -21:15 | 4.9 | 84 | 1009.0 | 296 | 4.1 | 43 | 10 | | | |
| 2012.8.8 | 08:23 -08:48 | 4.9 | 81 | 1007.6 | 238 | 2.1 | 38 | 11 | | | |
| 2012.8.9 | 14:40 -15:05 | 6.6 | 81 | 1003.9 | 129 | 6.5 | 146 | 50 | | | 12 |
| | 15:20 -15:49 | 7.0 | 78 | 1003.5 | 120 | 7.3 | 130 | 26 | 20 | | |
| 2012.8.10 | 00:15 -00:40 | 7.3 | 80 | 998.6 | 135 | 8.9 | 121 | 23 | | | |
| 2012.8.11 | 09:10 -09:35 | 6.2 | 94 | 997.0 | 303 | 3.3 | 128 | 50 | | | 10 |
| 2012.8.11 | 16:00 -16:25 | 4.1 | 92 | 1002.0 | 327 | 4.6 | 156 | 55 | | 6 | |
| 2012.8.12 | 15:25 -15:50 | 5.7 | 83 | 1006.8 | 132 | 6.9 | 100 | 15 | 32 | | |
| 2012.8.13 | 08:55 -09:20 | 5.3 | 81 | 1009.6 | 91 | 1.1 | 113 | 16 | | | |
| 2012.8.13 | 14:15 -14:40 | 4.5 | 90 | 1011.4 | 351 | 2.1 | 136 | 56 | | | 10 |
| 2012.8.14 | 09:50 -10:20 | 5.0 | 85 | 1019.7 | 351 | 2.3 | 134 | 24 | | | |
| 2012.8.14 | 15:12 -15:42 | 4.6 | 88 | 1020.5 | 117 | 2.6 | 121 | 26 | | | |
| 2012.8.14 | 21:17 -21:47 | 4.8 | 84 | 1020.7 | 276 | 5.4 | 178 | 56 | | 5 | |
| 2012.8.15 | 09:15 -09:45 | 5.8 | 73 | 1019.6 | 135 | 3.7 | 165 | 60 | | 6 | |
| 2012.8.15 | 15:00 -15:33 | 6.8 | 70 | 1018.9 | 270 | 3.3 | 80 | 11 | | | |
| 2012.8.17 | 9:00 -10:00 | 3.8 | 86 | 1017.1 | 116 | 0.3 | 30 | 15 | | | |
| 2012.8.17 | 14:50 -15:20 | 3.7 | 85 | 1015.7 | 109 | 2.2 | 42 | 16 | | | |
| 2012.8.21 | 15:05 -15:40 | 1.6 | 87 | 1003.7 | 314 | 6.8 | 46 | 18 | | | |
| 2012.8.22 | 08:55 -09:30 | 2.8 | 78 | 999.2 | 331 | 2.8 | 49 | 19 | | | |
| 2012.8.23 | 09:00 -09:40 | 3.4 | 64 | 998.0 | 136 | 6.9 | 21 | 9 | | | |
| 2012.8.23 | 20:35 -21:08 | 3.8 | 59 | 1002.0 | 138 | 6.3 | 25 | 9 | | | |

It would be helpful to provide a figure or two of the overall picture of aerosol composition, e.g., bar graphs or pie charts.

Response: Bulk aerosols were not determined in this study. We provide a summary of elemental compositions of individual particles (see figure below and Supplementary Fig. 3). These are based on elemental compositions of EDS. Based on the Figure S3, O, Na, S, Cl are most abundant elements in the arctic particles.

[Figure]

(a)                                              (b)

**Figure S3** Elemental compositions of individual particles from EDS spectra. Left: Average weight of elemental compositions derived from the EDS spectra (b) frequency of element occurring in individual particles.

There are percentages provided in the text, but showing the relative abundance of each particle type is pretty standard. Along relative abundance, the authors report that 29% of the particles were non-sea salt. What percentage were unclassified? What percentage is the "majority of NSS-particles"?

Response: Indeed, there are a few of unclassified particles. 63% of particles were identified as the sea salt particles, 29% particles were NSS-sulfate particles, and 10% particles were unclassified particles. The Figure as the referee requests was added in the main manuscript.

[Figure]

**Figure 3** Morphology and relative abundances of typical individual aerosol particles in the 21 analyzed samples.

As a result, it is not clear how important these particles are in general in the context of radiative impacts, given sea salt was what seemed like the dominant particle type and also largely affects scattering and SSA. In addition, to demonstrate that these particle types are important for the Arctic energy budget, they should show extinction properties for total aerosol (including sea salt) in Figure 6. I would think given the typical sizes for these types of particles (sulfate and soot) and reported abundance for this particular study, they would not affect the scattering cross sections relative to sea salt. The emphasis on the radiative impacts of these aerosol types is a large part of the manuscript, so their properties need to be presented in broader context. Are they important within the total aerosol population or not? This would be more relevant to the actual atmospheric implications.

Response: As suggested by the reviewer, the scattering cross section of sea salt particles is much higher than other aerosols. On the other hand, the purpose of this study is to show the potential role of BrC coating on sulfate particles on absorption. since sea salts do not absorb light, it is the soot and BrC that matters for the light absorption in the atmosphere within the whole aerosol population. does, and so does BrC. In the Arctic, it is the soot and BrC that has the potential to warm up the climate, even though their total absorption is likely to significantly lower than the absolute value of scattering.

The sizes of sulfate and soot particles are given in Figure 8. Our analysis cannot provide a clear information on the radiation balance but it did suggest the potential role BrC, which we know plays an important role in Arctic climate

In the revised manuscript, we revised Figure 8 and added the following (line 472 to 482)

"Figure 8c also shows that the single scattering albedos (SSAs) of individual particles are 0.92, 0.99, and 1 when assuming the BrC as strongly, moderately and non-absorbing (cases SSA1 to SSA3). These results suggest whether we consider organic coating as BrC may have a significant influence on the absorption properties of individual sulfate particles.

In this study, we expored the relationship between ACS of individual particles and particle diameters. Interestingly, Figure 8d shows that ACS of individual fine OM-coating sulfate particles increased following the increasing particle size. The result shows that the ACS can be enhanced following particle size growing and particle aging. In other word, OM-coating sulfate particles transported more longer distances and they might have stronger optical absorption in the Arctic air."

[Figure]

**Figure 8 Optical properties of Box-and-whisker plots showing optical parameters of all analysed particles assuming sulfate core and BrC shell (not considering soot cores in the particles)**. (a) Scattering cross section (b) Absorption cross section (c) Single scattering albedo. Top to bottom makers in the box-and-whisker represent max, 99%, 75%, mean, median, 25%, 1%, min values. (d) Absorption cross section along with particle diameter assuming strongly absorbing BrC and Moderate absorbing BrC as the particle OM coating.

There are several issues with the methods as presented. For instance, there is very little detail given on how the particles were classified under each technique, there are no errors or statistical analyses reported, and certain methods have very little specification detail (i.e., SEM is a very short paragraph). Regarding the samples storage, at 20% RH, I would assume all volatile and most semi-volatile species would evaporate, significantly altering particle shape, size, and composition. I will admit, these techniques are not my area of expertise, but the authors should at least comment on potential losses and caveats with this storage method. If there are significant losses of material, how representative are the analyzed particles of the total ambient aerosol population at the time of collection? I am skeptical the authors are comparing apples to apples by possible alteration of particles during storage.

Response: We add more specification detail of SEM in the paragraph.

SEM/ TEM/NanoSIMS all have to be analysed under vacuum so the volatile fractions will be lost during the analyses, but there is absolutely no other way to observe the shape of the particles from the ambient air. Laskina et al., EST, (2015) tested different methods and confirmed that the best way is settled in dry condition for further TEM or SEM analysis. Individual particle analyses offer the advantages of the mixing state and composition of individual particles.

The details about individual particle analysis have been reviewed by one previous paper (Li et al., JCP, 2016)

Laskina, O., Morris, H.S., Grandquist, J.R., Estillore, A.D., Stone, E.A., Grassian, V.H., Tivanski, A.V., 2015. Substrate-Deposited Sea Spray Aerosol Particles: Influence of Analytical Method, Substrate, and Storage Conditions on Particle Size, Phase, and Morphology. Environ. Sci. Tech. 49 (22), 13447-13453.

Li, W., Shao, L., Zhang, D., Ro, C.-U., Hu, M., Bi, X., Geng, H., Matsuki, A., Niu, H., Chen, J., 2016. A review of single aerosol particle studies in the atmosphere of East Asia: morphology, mixing state, source, and heterogeneous reactions. J. Clean. Prod. 112, Part 2, 1330-1349.

We've added the following the revised manuscript: L141-144

"Ambient laboratory conditions (17–23% RH and 19–21 °C) is effective at preserving individual hygroscopic aerosol particles and reducing changes that would alter samples and subsequent data interpretation (Laskina et al., 2015)."

L178-180"Particles examined by TEM were dry at the time of observation in the vacuum of the electron microscope. In our study, the effects of water and other semi-volatile organics were not considered as they evaporate in the vacuum."

Regarding the source analysis, there is very little detail given on the FLEXPART modeling and only a couple sentences on the results and discussion of the simulations. It is used to a very minimal extent and very generally summarized, even though Arctic aerosol sources can vary drastically day-by-day, and especially given possible local contributions. Figure 5 is very difficult to discern and glean any source information from it. Also, why are only these particular days shown?

Response: We revised the part related to the FLEXPART and added the sources around Arctic areas. Here we also added the Figure 1 that showing back trajectory of air mass during the sampling periods.

The reason we shosed the FLEPART at the specific days is that we did FLEPART based on the preliminary works including back trajectories of each sampling and TEM study. During these days, We found abundant sulfate and soot in samples in 9-15, Aug (Table S1). This is the reason that we planned to the FLEXPART modeling.

Figure 2 could not give direct source information but they can provide potential source locations. Here we added two Figures in the supplemental which provide the emission intensity in the Arctic area.

[Figure]

**Figure S2** OC (a) and SO$_2$ (b) Emission intensity in Arctic area and 24h back trajectories on August 11, 12, 14, and 15, 2012.

[Figure]

**Figure 1** 72 h back trajectories of air masses at 500m over Arctic Yellow River Station in Svalbard during 3–26 August 2012, and arriving time was set according to the sampling time

There is no background on previous relevant studies conducted at the study location, even though there is a long-term monitoring station with aerosol measurements at Ny-Ålesund (https://www.esrl.noaa.gov/psd/iasoa/stations/nyalesund). It is not the same exact location as the Chinese site, but close enough to at least use those routine, publically-available measurements to provide some broader spatial and temporal context.

Response: this is now being revised. We've added L127-130

"The sampling site is about 2 km far away from the Zeppelin observatory station (78.9N 11.88E) running by the Ny-Ålesund Science Managers Committee (https://www.esrl.noaa.gov/psd/iasoa/stations/nyalesund). Two to three samples were regularly collected at 9:00, 16:00, 21:00 (local time) of each day, with a total of 46 samples during 7-23 August, 2012."

There is only basic mentioning of biogenic VOCs, but none of biogenic or biological aerosol. The Arctic summer, especially in remote coastal sites, is largely affected by gases and aerosol from primary productivity due to the availability of sunlight and open water. There is no discussion on if the OM is biogenic/biological, and in general, the definition of OM is vague.

Response: The methodology we use cannot identify the sources of the SOAs, whether biogenic or non-biogenic. In the arctic atmosphere, Leck and Svensson (2015) found some biogenic aerosols like gel-aggregate containing bacterium in ultrafine particles. In our study, we didn't collect ultrafine particles using the sampler.

We've added the following to line L405-408

"Similarly, besides the OM coating in the Arctic particles, Leck and Svensson (2015) found some biogenic aerosols like gel-aggregate containing bacterium in ultrafine particles. However, we didn't find any gel-like particles in the samples because our sampler had very low efficiency for ultrafine particles."

Technical corrections:

There are many typos, grammatical issues, and a lack of necessary explanation (e.g., the 3 sets of bars in Figure 6, SEM a short paragraph with no numbers, etc.).

Response: Thanks. We revised them.

References:

Buseck, P.R., Posfai, M.: Airborne minerals and related aerosol particles: Effects on climate and the environment, P. Natl. Acad. Sci. USA, 96 (7), 3372-3379, 1999.

Chi, J.W., Li, W.J., Zhang, D.Z., Zhang, J.C., Lin, Y.T., Shen, X.J., Sun, J.Y., Chen, J.M., Zhang, X.Y., Zhang, Y.M., Wang, W.X.: Sea salt aerosols as a reactive surface for inorganic and organic acidic gases in the Arctic troposphere, Atmos. Chem. Phys., 15 (19), 11341-11353, 2015.

Laskina, O., Morris, H.S., Grandquist, J.R., Estillore, A.D., Stone, E.A., Grassian, V.H., Tivanski, A.V.: Substrate-Deposited Sea Spray Aerosol Particles: Influence of Analytical Method, Substrate, and Storage Conditions on Particle Size, Phase, and Morphology, Environ. Sci. Tech., 49 (22), 13447-13453, 2015.

Leck, C., Svensson, E.: Importance of aerosol composition and mixing state for cloud droplet activation over the Arctic pack ice in summer, Atmos. Chem. Phys., 15 (5), 2545-2568, 2015.

Maahn, M., de Boer, G., Creamean, J.M., Feingold, G., McFarquhar, G.M., Wu, W., Mei, F.: The observed influence of local anthropogenic pollution on northern Alaskan cloud properties, Atmos. Chem. Phys., 17 (23), 14709-14726, 2017.

**General Response: We thank the reviewer for your helpful comments. We have addressed all comments and provided point by point response below. The revised manuscript is presented in below.**

**Response to the referee#2' comments**

Overall Comments: The authors use multiple measures to investigate the physical properties of non-sea salt aerosols collected in the Arctic. They qualitatively describe the mixing state of organic, soot, and sulfate aerosols. Although the core data set of this manuscript is informative, as is, the presentation and discussion are confusing and possibly misleading. No large scale descriptive statistics are provided, the methods and presentation are often unclear and/or redundant, the environmental context is lacking, and the optical conclusions are not well explained. This manuscript would benefit from additional analysis, clarification of methods, adding proper context to results, and careful grammar review.

Response: We carefully addressed all of the questions below. We attempted to re-write the manuscript. However, we are not able to address non-specific questions.

Unclear Sample Selection: In multiple sections of this paper the authors discuss a subset of their samples without explaining why they were chosen. This detracts from their discussion and leaves the reader questioning what is not being discussed. Anytime a subset of samples is chosen or used, an explanation needs to be provided explaining why these are the best samples for that specific analysis or investigation. A few example occurrences can be found in lines 275, 281, 118, 190.

Response: Thank you for good comments. We received some comments before and revised the manuscript several rounds. After that, we added Figure S1 showing what samples were analyzed by different instruments. We removed some number which confused for the readers and rephrased some sentences in the Mehods section.

To SEM and NanoSIMS analysis after TEM observations, we need to select better samples. These samples should keep good conditions of carbon film because some samples have broken carbon film during the sampling period. In the NanoSIMS and SEM analysis, these samples need to be sticked on the sample stage, they could not be recycled again.

Please see response to reviewer 1, comments 2-4.

Environmental Context: The only environmental context explored in this text is the back trajectories provided for 4 of the sampling days. Whenever field data is discussed, external variables such as temperature, humidity, and back trajectories can help explain variability observed in the data. To improve this work, the context of the sampling should be explored as a driving factor between differences in particle characteristics.

Response: We added the RH, T, P in the table S1. These meteorological data were recorded during the sampling period. Here we did back trajectories of each sampling day as shown in Figure 1.

We added the following to the revised manuscript (line 144 to 146)

"The sample information such as local sampling date and time and meteorological conditions (e.g., temperature (T), relative humidity (RH), pressure (P), wind direction (WD), wind speed (WS)) were listed in Table S1."

[Figure]

**Figure 1** 72 h back trajectories of air masses at 500m over Arctic Yellow River Station in Svalbard during 3–26 August 2012, and arriving time was set according to the sampling time

2.1 I am also not convinced that the daily averaged back trajectory calculations are meaningful for these samples. (I'm also not sure the trajectories were daily averaged as the methods section just said they were calculated for a given day, not the frequency of times or given time on the day.) Because the sample length was between 20 minutes and 2 hours, a back trajectory occurring during the actual sampling time for all samples would be more appropriate.

Response: We did choose the mid of the sampling time as the end point for the back trajectory analysis (see Figure 1)

Optical Property Calculations: The authors attempt to estimate the potential radiative forcing implications of their results using a core-shell model and Mie theory. Although this is a potentially meaningful result, the methodology needs to be discussed further and improved.

Response: We added more description in the Method section.

L263-271"BHCOAT Mie code by Bohren and Huffman (1983) was used to calculate the optical properties, including scattering cross section (SCS), absorption cross section (ACS), and single scattering albedo (SSA), assuming a core-shell structure. We firstly calculated these parameters assuming a sulfate core and OM shell structure only (ignoring some of the particles that contain soot core). Because the Mie code only can calculate the core-shell structure or homogeneous models, we assume sulfate as a core and OM as a shell in individual particle to build the core-shell model. Based on the core-shell standard mode (Li et al., 2016), we can calculate optical properties of individual internally mixed particles."

3.1 When performing Mie theory calculations the input size distribution characteristics and wavelength dependence (angstrom exponent) have dramatic effects on the results. These data need to be reported. The authors state that a single size distribution is used. How was this size distribution established? Could you run your analysis over all of the size distributions you observed to estimate the variability?

Response: Size distributions of particles are measured by microscopes These measurements can help us to evaluate sulfate and OM volume and then we made the Figure 8d. Once we obtained these basic data, we can input all the data to calculate optical properties of individual particles. Here the result is different from the bulk online optical properties. As the comments, we did correction between optical absorption cross section (ACS) and particle diameter assuming strongly absorbing BrC and Moderate absorbing BrC as the particle OM coating. Interestingly, we found that ACS increase along with the particle diameter increase through one nonlinearity showed as below.

In this study, it is not necessary to consider wavelength dependence because lots of work how different size of aerosol particles interacted with the different wavelength. We just did pilot cases to confirm how BrC influence the optical absorption of sulfate particles based on our TEM data (Figure 8). The purpose of this study draws attention for BrC in the Arctic air.

[Figure]

**Figure 8 Optical properties of Box-and-whisker plots showing optical parameters of all**

**analysed particles assuming sulfate core and BrC shell (not considering soot cores in the particles)**. (a) Scattering cross section (b) Absorption cross section (c) Single scattering albedo. Top to bottom makers in the box-and-whisker represent max, 99%, 75%, mean, median, 25%, 1%, min values. (d) Absorption cross section along with particle diameter assuming strongly absorbing BrC and Moderate absorbing BrC as the particle OM coating.

3.2 The way the authors include soot in their calculations is not appropriate, as soot is treated as a core with a mixed sulfate-OM shell. This is in direct conflict with their observations that soot is only observed to be associated with the OM shell and sulfate is always the core. It would be better not to include a soot calculation than include a misleading one.

Response: We deleted the soot part here. The updated data is shown in Figure 8

3.3 The relevancy of the refractive index (RI) choice needs to be discussed further. The authors use an RI from biomass burning brown carbon as their slightly absorbing case but frequently state that they believe the OM to be secondary. If that is the case, their estimations are likely too high for all cases but the non-absorbing case. The authors should include refractive indexes from secondary organic aerosol brown carbon to get a more realistic answer. These are generally much lower than the RI values that they used. Additionally, the scattering component chosen (1.65) is relatively high and not explained. Please explain why this value was chosen as it is especially important in core-shell cases.

Response: There are many different RIs from the laboratory experiments. For example, Jiang et al., (2019) suggested the measured RIs on different organic species. The RI is dependence on the wavelength. Fortunately, we just want to test how OM coating influence the sulfate particles. Based on the relationship between RIs and wavelength (Jiang et al., 2019; Feng et al., 2013), the RI doesn't influence our conclusions in this study.

Reference added: Jiang, H., A. L. Frie, A. Lavi, J. Y. Chen, H. Zhang, R. Bahreini, and Y.-H. Lin (2019), Brown Carbon Formation from Nighttime Chemistry of Unsaturated Heterocyclic Volatile Organic Compounds, Environmental Science & Technology Letters, DOI: 10.1021/acs.estlett.1029b00017.

Here we chose the RI reported in Feng et al., 2013. Through the comparisons between strongly absorbing and moderately absorbing, non-absorbing BrC, we generally knew how BrC coating could influence optical properties of sulfate particles. Although our current study could not get a more realistic answer, but the result suggests that BrC has a potentially role to play in light absorption properties of aerosols Arctic. The conclusion warrants further study in the Arctic area. Our results from the TEM can provide solid evidence about the mixing state of sulfate and other aerosol species. As suggested by the referee, we added a sentence to explain why we selected the 550 nm:

In the context:

"Although the refractive index has dependence on the wavelength between 350-870 nm, we tried to select the 550 nm as a case to test how OM coating influence sulfate particles in Arctic air."

3.4 Stemming from the above comment, the relevancy of each refractive index case needs to be discussed in the context of the Arctic. Which case seems the most likely?

Response: We can't make this judgement because there is no data to show this. We showed that this is potentially important to consider, which warrants further study.

3.5 The authors use m=1.55 for sulfate. RI values have wavelength dependence, was this included in any way?

Response: Thanks. It does. We corrected the writing here.

Line 256 to 257: "The refractive index used for the non-light-absorbing sulfate component was set to m=1.55 at 550 nm (Seinfeld and Pandis, 2006)."

3.6 The authors average the absorption cross section on a per particle basis. This isn't meaningful since observed absorption cross sections will depend on the whole aerosol population (and the size of the particle). The authors should calculate an absorption cross section for the ambient aerosol concentration during their sampling and compare it to other absorption observations.

Response: Our methods are not able to give the ambient aerosol concentration. We tested how the absorbing OM influenced optical properties of sulfate in the Arctic based our individual particle measurements. These results showed that this is potentially important to consider, which warrants further study.

Redundancy and Clarity: The writing of this paper needs improvement. There are multiple times when the authors restate the same point twice or fail to introduce a topic before discussing their results. An example of redundancy can be observed when Copper TEM grids are introduced in lines 120, 140, and 156. An example of an improper discussion occurs in the discussion of satellite particles, which are introduced in line 280 with no context or explanation. This forces the reader to infer what the authors mean by satellite particles, possibly leading them to wrong conclusions. To improve the manuscript, I recommend careful reconsideration of the presentation of the data, with special consideration to avoiding redundancy and ensuring the appropriate context is present.

Response: We carefully revised them. Please see our replies as below associated with your specific comments.

4.1 There are multiple points in the methods, a few of which I've included in the specific comments, that are unclear or confusing. The reader needs to be able to understand exactly which analysis was performed on which filters for how many particles if they are to believe your result.

Response: We made one major revision in this section. We added the Table S1 to make specific information.

Sample Information: The only summary of the total data set is provided in figure S6 and this figure states that only 3 samples (of 46 collected and 21 analyzed). No explanation is given as to why is summary is so limited given the authors have EDS data (which they used for classification) of 20-30 particles over 21 samples (at least 400 data points). All the of following discussions only make sense if they are provided in the context of overall sample composition.

Response: The Referee is probably referring to Figure S4 rather than Figure S6. It is a spelling mistake, which we have corrected. At the revised ms, the Figure was moved to the main ms as Figure 1.

Specific Comments:

There are many grammatical errors and redundancies that I have not addressed below.

Response: We carefully improved English writing in the manuscript.

The phrase internal mixing is used throughout the manuscript without an explicit definition. An explanation of what exactly you mean when you say something is "internally mixed" would improve the manuscript

Response: We added the following to the text

Add an explanation the first time you write it, e.g., line 88

"Internal mixing means that a single particle simultaneously contains two or more types of aerosol components (Li et al., 2016)."

Line 55: Change "Artic" to "the Artic"

Response: Corrected

Line 60: Change "nature" to "natural"

Response: Corrected

Line 64-66: Restructure this sentence for clarity. Treat the percentages in a consistent way as to not confuse the reader. For example, this sentence could be changed to:" For example, Winger et al.(2017) showed most Arctic BC is sourced from domestic activities (35%) and transportation (38%), with only minor contributions from gas flaring (6%), power plants (9%), and open fires (12%)"

Response: Corrected

Line 72: I'm not sure exactly how this sentence fits in with the brown carbon theme of this paragraph. Are these compounds commonly found in brown carbon or organic aerosols in general? Please add some context.

Response: Thank you very much. We move the positions of this sentence and add more description here.

L73-78: "Accumulation of secondary organic aerosol, a significant fraction of the new particles grow to sizes that are active in cloud droplet formation in the Arctic (Abbatt et al., 2019). More than 100 organic species were detected in the Arctic aerosols and polyacids were found to be the most abundant compound class, followed by phthalates, aromatic acids, fatty acids, fatty alcohols, sugars/sugar alcohols, and n-alkanes (Fu et al., 2008)."

Line 84: replace "were" with "have been"

Response: Corrected

Line 96-68: This sentence is confusing. Please rewrite it more concisely and clearly.

Response: Revised as the following (line 106 to 109)

"The poor understanding on mixing state of BC and BrC in individual particles will prevent the further simulation of atmospheric climate and aerosol-cloud interaction in the Arctic through the current atmospheric models (Browse et al., 2013; Samset et al., 2014; Zanatta et al., 2018)."

Line 99-100: Change "collected on 7 to 23 August, 2012 in the Arctic." to "collected in the Artic between August 7th and 23rd, 2012."
Response: Corrected.

Line 104: replace "on substrate" with "on a substrate"
Response: deleted the word here

Line 118: Change "samples between 7 and 23 August, 2012." to "samples collected between August 7th and 23rd, 2012."
Response: Corrected

Line 119: Replace "analyzed for TEM analysis" with "analyzed with TEM"
Response: Revised

114-131: Restructure your sampling section. As written the reader may think that you sampled with 2 separate samplers an individual particle sampler and cascade impactor. After reading the paper, there is only one sampler. This confusion can be remedied by introducing the cascade impactor earlier.
Response: Revised the part as below (line 133 to 136)
"A sampler containing a single-stage impactor with a 0.5-mm-diameter jet nozzle (Genstar Electronic Technology, China) was used to collect individual particles by the air flow rate at 1.5 l min-1. Aerosol particles were collected onto copper TEM grids coated with carbon film."

122-123: Add the top size cutoff for this sampler.
Response: Added

138: replace "within a" with "for a"
Response: Corrected

148: 2002 particles examined over all the samples, or in a specific filter?
Response: Added the Table S1 in the sentence. Table S1 can show how many particles we analyzed in each sample.

151: Clarify this sentence. Do you mean to say you ", we only checked elemental compositions of 20-30 particles" in each sample?
Response: We added more detailed description (line 165 to 180)
"In the clean Arctic air, there are simply particle types including sea salt, sulfate, soot, and OM. Because soot particles have chain-like aggregation, it is not necessary to check their elemental composition. Sea salt particles display spherical or square shapes and are stable under the electron beam in TEM but sulfate particles are spherical but flats on the substrate and produce unstable bubble under the electron beam (Buseck and Posfai, 1999; Chi et al., 2015). TEM observations also can clearly identify sulfate particles or sulfate with OM coating. Therefore, we can easily identify Arctic particle types based on their morphology. Because of the time-consuming in the experiment, it is not necessary to frequently check elemental composition of the same particle type. For the data statistic in this study, we randomly checked elemental composition of 20-30 particles in each sample (Table S1). EDS spectra of 575 particles were manually selected and saved in the computer for elemental composition analysis. Particles examined by TEM were dry at the time of observation in the vacuum of the electron microscope. In our study, the effects of water and other semi-volatile organics were not considered as they evaporate in the vacuum."

155-156: The statement about Cu is redundant. This has already been stated in line 120 and 140.
Response: Corrected:

156-157: What is the difference between what is stated here and what is stated in lines 147-149?
Response: In lines 147-149 descript how to analyze particles on the substrate using TEM/EDS.
In 156-157, we used scanning TEM. The method is one other function in the TEM which can give element profile along with the one line or elemental mapping in the targeted individual particle. We added more description here and tell the reader why we did the STEM here.

The following paragraph was revised to (line 181 to 188)
" Elemental mapping and line profile of individual aerosol particles were obtained from the EDX scanning operation mode of TEM (STEM). The STEM information can clearly display elemental distribution in the targeted individual particles which cannot be provided by the above EDS examination. Based on preliminary individual analysis, we further chose the typical samples containing abundant sulfate with OM coating for the STEM analysis. The high-resolution details of elemental distribution in individual particles can further prove the details of the mixing structure of sulfate and OM in individual particles."

162-163: Replace "is the image analysis platform…" with "is an image analysis platform".
Response: Corrected

177: Replace "Organic Matters" with "Organic Matter"
Response: Corrected

Line 184: Replace "TEM grids was" with either "The TEM grid was" or "TEM grids were"
Response: Corrected

Line 192-194: Incomplete sentence
Response: Corrected
"AFM with a digital nanoscope IIIa instrument operating in the tapping mode was used to observe surface morphology of individual aerosol particles and measure particle thickness."

Line 194: I stopped making basic grammar and structure critiques at this point.

Response: Thank you very much. We carefully checked them and improved the English writing. Please see the red markers in the revised ms.

Line 222: Does treating this as a core-shell system with BC in the middle and sulfate and OM mixed on the outside have any basis? You've indicated that you have soot inclusions on the outside of predominantly sulfate particles, so why would soot be on the inside?
Response: We've deleted the part with soot – see response in comment above.

Line 224: This sentence reads as if you've calculated the refractive index of the particles. Did you measure the optical properties of these particles?
Response: We didn't measure their optical properties. We deleted this sentence and revised the part.

Line228: "In this study". It is unclear whether this is referring to your previous work or this manuscript.
Response: corrected

Line 253: Is it possible that coagulation of primary organic particles and S-rich particles could have led to the formation of organic coatings? Are you sure assumption that all organic coatings are secondary valid?
Response: The possibility could happen in the polluted air due to high particle number. As our previous study, Chi et al., (2017) determined sea salt particles collected in the Arctic air. We didn't observe the association of sulfate and sea salts particles, although we found many aged sea salt particles. The observations again suggest the particle coagulation is unlikely to be important in the Arctic air.

Line 272: Can you say a percentage of NSS particles that are S-rich with an OM coating? Or a percentage of S-rich particles that have a coating? This would strengthen the paper if an actual number was given.
Response: No, we cannot say NSS particles are S-rich with OM coating particles. Figure S4 shows 39% by number of all the analyzed particles were NSS-particles but 29% particles contain sulfate. Here we found 73% of the analyzed NSS-particles are S-rich with OM coating. We've changed this line 310 to 312: "Here we focused on S-rich, soot, and OM particles as the major non-sea salt particle (NSS-particle, 39±5%) in the analyzed samples, which are approximately 29±7% of 2002 particles (Figure 3). "
L323-324: "A majority of 781 analyzed NSS-particles (74% by particle number) have a sulfate core and OM coating (Figures 4 and 5)."

Line 274-276: Are these specific samples special or is there something that you think may have caused the low frequency of soot inclusions? If so, please explain why.
Response: We only found some fresh soot particles in three samples out of the 21 samples. We found many sulfate with soot inclusions. Previous works showed that BC is very low in the Arctic. We've changed this (line 326 to 333).

"The mixing structure is different from our previous findings in polluted air that soot is normally mixed with sulfate instead of OM coating (Li et al., 2016). Moreover, we noticed that a few chain-like soot aggregates (1.3% in all analyzed particles) (Figure S5) only occurred in three samples during the sampling period (Table S1). Considering the remoteness of the sampling site, such fresh soot particles are likely to be of local origin, including shipping and flaring (Gilgen et al., 2018; Peters et al., 2011). Indeed, we found a few of ships moving in Arctic Ocean during these days from the Ny-Ålesund town."

Line 277: This statement needs to be better supported. Just because a site is remote does not mean particles are local in origin. If this is supported by your trajectory calculations, mention them here. Also, don't the soot inclusions also imply that perhaps the OM is not secondary in nature? Soot is 100% primary and often co-emitted with primary OM, so if there is soot associated with OM coatings the soot itself is primary and so possibly some of the OM is primary as well.

Response: We found fresh soot particles in Figure 6. Indeed, TEM image shows that the soot particles have very thin OM coating.

It is true that soot is often co-emitted with primary OM. However, there are plenty of evidence to show that the primary soot aggregate particles are not heavily mixed with OM and in particularly not coated with thick OM (Wang et al., 2017). Furthermore, It is well known that SOA is dominant in the remote air (Jimenez et al., 2009).

Line 280: You need to define what satellite particles are; you have not discussed or defined them previously. Are they simply splatter of liquid portions of the particle when the particle is collected?

Response: Thanks. We define it.

L334-337 "TEM observations showed that some sulfate particles had unique morphology that a sulfate particle was surrounded by some smaller particles (Figure 5a). They are often called "satellite" particles as they were distributed from the central particles when impacted on the substrate during sample collection."

Line 280-281: is there a reason why satellite particles would have been observed on these days but not other days?

Response: The samples collected during 9-15, August contained abundant sulfate particles, OM, and soot particles (Table S1). Also, Figure 2 shows that air masses from North American during these days were mainly dominant. The reasons should depend on their air masses. Although the mechanism of this point is interesting, we could not give specific evidence under the current data.

Line 281: This is misleading and implies you performed the NanoSIMS analysis on 11 samples. In the methods section, it says only two samples were analyzed with NanoSIMS.

Response: Corrected

L339-342: "NanoSIMS analysis further provided more information that the satellite particles selected from the samples (Table S1) have strong 32S- (Figure 6a, c) and 16O- signals (Figure 6d) as well as weak 12C14N- signals (Figure 6a, b)."

Line 287-289: This is misleading. It reads as if you have done molecular characterization of the organic matter.
Response: Corrected
L346-347 "Indeed, Fu et al. (2008) found that polyacids are the most abundant organic compounds, followed by phthalates, aromatic acids, and fatty acids in Arctic aerosol particles."

Line 314: You back trajectories are only for specific days, be transparent about this in the Discussion
Response: Corrected. Add one Figure S2

Line 317-318: This conflicts with your earlier comment that most BC should be local.
Response: The result does not conflict with earlier comments. We only pointed out the fresh soot particles (only 1.3%) may come from local shipping or combustion activitgies. It doesn't mean most BC were from local sources. In this study, we did not determine where the BC comes from.

Line 326-342: This discussion reads like a list of facts, but why they are all relevant is not always stated. Explain why each observation is important and how it adds context to your results.
Response: We carefully revised the part as below
L338-408: "The sulfate core-OM shell structure observed in the Arctic summer atmosphere is similar to those in the background or rural air in other places (Li et al., 2016; Moffet et al., 2013). Based on the images from electron microscopies, we can infer that OM coating thickness in the arctic atmosphere was comparable with them in rural places but higher than them in urban places. During the transports, organic coatings on sulfates were considered as the secondary organic aerosols and their masses increase following particle aging and growth (Li et al., 2016; Moffet et al., 2013; Sierau et al., 2014). Figures 1 and 2 show that most of particles in the air masses transported long distance from North American. The result indicates that these long-range transportation of secondary sulfate particles have enough time to experience the possible atmospheric heterogeneous reactions on particle surfaces or cloud processes in the Arctic air. Similarly, Moffet et al. (2013) found that soot inclusions occurred in OM coating when OM coating on sulfates built up through photochemical activity and pollution buildup the Sacramento urban plume aged. On the other hand, the sulfate/OM particles with soot inclusions are probably formed in a similar way as those found elsewhere (Li et al., 2016) – e.g., soot particles may have acted as nuclei for secondary sulfate or organic uptake during their transports (Riemer et al., 2009). Similarly, besides the OM coating in the Arctic particles, Leck and Svensson (2015) found some biogenic aerosols like gel-aggregate containing bacterium in ultrafine particles. However, we didn't find any gel-like particles in the samples because our sampler had very low efficiency for ultrafine particles."

Line 343: Why is dry included here? Are there also wet particles that you have not discussed?

Response: Deleted the "Dry" here

Line 344-347: Explain LLPS in simpler terms and why it's important.

Response: Revised the part.

L409-428: "TEM images show that most of the internally mixed sulfate particles display sulfate core and OM coating on the substrate (Figures 4a and 5b, c). The sulfate and OM separation in individual particles were defined by You et al. (2012) as liquid-liquid phase separation (LLPS). Concerning the knowledges of the LLPS can better understand particle hygroscopicity, heterogeneous reactions between reactive gases on particle surface, and organic aging (You et al., 2012). They also reported that the LLPS can reflect the O:C ratio in the OM, which is roughly ≤ 0.5. In this study, we did observe the LLPS in almost all the fine sulfate particles, which indicates that the secondary OM in the coating might be not highly aged. Therefore, we speculate that the thick OM coatings were consistently built up during the long range transport of sulfate particles and part of secondary OM in the coating likely formed in Arctic area. Indeed, some studies reported that there are various sources of organic precursors during the Arctic area, such as biogenic VOCs from ice melting and open water (Dall´Osto et al., 2017) and anthropogenic VOCs from shipping emissions in summertime (Gilgen et al., 2018). The dependence of OM volume on particle size (Figure S6) suggests that the suspended sulfate particles are initially important surface for secondary OM formation. Moreover, the common OM coating on sulfate particles indicates that secondary OM as the surfaces of fine particles might govern the possible heterogeneous reactions between reactive gases and sulfate particles in the Arctic air."

Line 346: It's unclear what 90% is referring to in this statement.

Response: We deleted this sentence here

Line 348: There's likely a complex relationship between phase state, oxidation state, and humidity. This needs to be investigated and explained further if statements about aerosol age are going to be made. Additionally, shouldn't you see a variety of ages of aerosol in your samples? Showing contrast between aged and unaged particles would be interesting and convincing.

Response: We re-wrote this part. We pointed out that the implications including LLSP in the Arctic air. Almost all particles we observed the aged. We do not have a parameter to define the age of the aerosol population and we are unable to answer the question clearly.

Line 376-378: This is circular reasoning because 12C14N- was what you used to identify OM so of course it was observed in the OM coatings. I don't think this data set is appropriate to make conclusions about the N content of OM coatings. That said, if you were able to calculate the mass concentration of N in the coatings with NanoSIMS that might give you a better indication of the BrC potential of the OM.

Response: It is impossible to calculate the N concentration in the coatings based on the NanoSIMS data here. The literature as the below proved that CN- can indicate N-containing organic matter.

Herrmann, A. M., K. Ritz, N. Nunan, P. L. Clode, J. Pett-Ridge, M. R. Kilburn, D. V. Murphy, A. G.

O'Donnell, and E. A. Stockdale (2007), Nano-scale secondary ion mass spectrometry — A new analytical tool in biogeochemistry and soil ecology: A review article, Soil Biology and Biochemistry, 39(8), 1835-1850.

Line 390-393: The average absorption cross-section is reported on a particle basis. This would be much more meaningful if it was extrapolated to environmental conditions. Because you've sampled from the atmosphere, you should be able to approximate particle concentrations, correct? Is so, you could back calculate this to an actual atmospheric absorption contribution and compare it to expected absorption from other species and measurements. This would be significantly more meaningful.

Response: Yes, we agree it would be useful to extrapolate to environmental conditions but as stated by the reviewers, we do not want to over-interpret our results. We cannot extrapolate because we do not have ambient particle concentration data. One single paper will not be able to answer all questions and we do suggest for future work to answer these questions.

Line 413, 260: 29% number should include a standard deviation.
Response: added

Line 410: The last section shouldn't simply repeat what was stated in the above sections, but instead present the data in additional context and discuss the implications.
Response: We made major revision in the section. Please see the revised red words.

References:

[revised manuscript text omitted]

---

## Referee Report (RR1)

Yu et al. present findings from detailed compositional measurements of Arctic aerosol in Svalbard during August 2012. The comprehensive combination of analytical techniques employed is incredibly useful for detailed aerosol characterization and source apportionment, in addition to the use of air mass trajectory analyses. While there is obvious importance of conducting detailed physiochemical characterizations of Arctic aerosol in terms of their radiative impacts and subsequent indirect effects on frozen surfaces, there are several issues with the manuscript by Yu et al. that would need to be addressed prior to publication, as discussed in more detail below.

**General comments:**

Generally, the introduction could use some restructuring. The climate impacts paragraph could be expanded upon to include more details on specific aerosol types and how they affect the radiative budget and cloud microphysics, providing the motivation for why detailed characterization of aerosols in the Arctic is important – i.e., different aerosols have different effects. This is especially so since the last sentence of the introduction explicitly states that results are discussed in the context of aerosol-radiation and aerosol-cloud interactions. Thus, background material on these effects is needed. The information on lines 92-100 should be located sooner in the introduction as it provides a nice general statement of aerosol studies in the Arctic. There is no background on previous relevant studies conducted at the study location, even though there is a long-term monitoring station with aerosol measurements at Ny-Ålesund (https://www.esrl.noaa.gov/psd/iasoa/stations/nyalesund). It is not the same exact location as the Chinese site, but close enough to at least use those routine, publically-available measurements to provide some broader spatial and temporal context. Then, the discussion on the background on sources of BC, OM, BrC, and sulfate and the various mixing mechanisms can reside.

Laskina et al. (2015) states, "Typically, SSA particles are deposited wet and, if possible, samples used for single-particle analysis should be stored at or near conditions at which they were collected in order to avoid dehydration." I assume the RH was much higher during collection, so can the authors comment on what might change between collection and storage under the different RH conditions? Also, Laskina et al. is focused on sea spray aerosol, but the authors conclude that continental sources were a major contributor (lines 377-379), so what about loss of semivolatile organics or sulfate during low RH storage? This possible caveats should be discussed in more detail than they currently are.

"Sulfate" particles is used intermittently throughout, perhaps "sulfate-containing" is more appropriate.

**Specific comments:**

Title: Because this work is only conducted at one location during one month, extending it to "summer Arctic" seems like a stretch as aerosol populations can vary significantly from terrestrial to the high Arctic and can also change from early to late summer. Perhaps the authors should consider changing the title to something like, "Organic coatings on sulfate and soot particles during late summer in the Svalbard Archipelago".

Lines 33-34: State the motivation for the particular focus on these particle types, so that this focus is justified.

Lines 34-35: State percentage of OM coated NSS-sulfate here to provide some quantification to "commonly coated".

Lines 52-53: Regional pollutants and local natural aerosol production can also affect sea ice albedo, especially in the summer when midlatitude transport is not as frequent relative to the winter/spring Arctic Haze season.

Lines 57-63: This part is very BC focused and should be included in the second paragraph.

Lines 144-146: There are no details provided on where these meteorological data came from or what instrumentation was used to measure the mentioned parameters. Although, the data are not presented anywhere so perhaps this information is not relevant in include at all.

Line 193: Define the size of "fine" and quantify "abundant".

Lines 200-201: "good consistent property" is vague.

Section 2.4: Separate SEM and AFM into separate sections – not sure why these specifically are combined.

Line 275: Why only 500 m?

Lines 276-279: This belongs in the results section.

Lines 290-292: Why were different days evaluated for HYSPLIT versus FLEXPART (3 versus 10 day, respectively)?

Lines 304-306: Provide a citation for this statement.

Lines 310-312: 39% for the s-rich + soot + OM out of all NSS-particles? How was NSS-sulfate and OM determined?

Lines 332-333: How was this documented? This statement is a bit vague.

Figure 1: The date are very small and it is difficult to tell if there is any sort of time evolution in the air mass sources throughout August. Perhaps the authors could either color the lines by date or create multiple panels (e.g., 1 per week of trajectories).

Figure 3: I assume this is from TEM/EDX? Why are soot and OM not shown? Also, can the authors show a time series in addition to the diameter relative fraction? That might provide some insight into temporal changes due to air mass origin.

Table S1: There is a substantial amount of referencing to Table S1. While I am glad to see the authors have include sample and particle numbers for each analytical technique, jumping back and forth between the manuscript and SI is tedious. I suggest the table be moved to the main manuscript.

**Technical corrections:**

There are many typos and grammatical issues that need to be fixed throughout the manuscript.

Line 32: "mixing state properties"

---

## Author Response (AR2)

Dear Dr. Wang

We really appreciated that you handle the manuscript reviewing processes.

We have carefully considered all comments and re-wrote the manuscript substantially.

(1) We restructure the introduction section and add three new paragraphs to introduce each particle type and their impacts in the Arctic air.

(2) We replaced the Figure 1 as the referee's comments.

(3) We edited the English through the whole manuscript indicated by the red color.

(4) We seriously revised the whole manuscript with the referee' comments. Our responses were uploaded with the revised the manuscript.

The point to point responses and the revised manuscript with track markers were listed as below. We really appreciated the referee's comments which significantly improve the quality of this manuscript. We hope that the current version can be accepted and published in the ACP.

Sincerely

Weijun Li & Zongbo Shi

General comment: While there is obvious importance of conducting detailed physiochemical characterizations of Arctic aerosol in terms of their radiative impacts and subsequent indirect effects on frozen surfaces, there are several issues with the manuscript by Yu et al. that would need to be addressed prior to publication, as discussed in more detail below.

Response: We carefully revised the manuscript considering all comments.

Comment 1: Generally, the introduction could use some restructuring. The climate impacts paragraph could be expanded upon to include more details on specific aerosol types and how they affect the radiative budget and cloud microphysics, providing the motivation for why detailed characterization of aerosols in the Arctic is important – i.e., different aerosols have different effects. This is especially so since the last sentence of the introduction explicitly states that results are discussed in the context of aerosol-radiation and aerosol-cloud interactions. Thus, background material on these effects is needed. The information on lines 92-100 should be located sooner in the introduction as it provides a nice general statement of aerosol studies in the Arctic.

Response: We restructured the introduction part. We moved BC part into second paragraph and line 92-100 into the first paragraph. We also added three paragraphs to summarize the different aerosols including sea salts, mineral, and sulfate, that affect the climate and clouds. In this study, we focus on the sulfate and OM.

Based on the comments, we added more details about the motivation

Line 50-61 "Spatial and temporal variations of aerosol composition, size distribution, and sources of Arctic aerosols have been studied extensively in numerous ground-based, ship, airborne observations, and various atmospheric models (Brock et al., 2011; Burkart et al., 2017; Chang et al., 2011; Dall Osto et al., 2017; Fu et al., 2008; Hara et al., 2003; Hegg et al., 2010; Iziomon et al., 2006; Karl et al., 2013; Lathem et al., 2013; Leck and Bigg, 2008; Leck and Svensson, 2015; Moore et al., 2011; Raatikainen et al., 2015; Wörnschimmel et al., 2013; Winiger et al., 2017; Yang et al., 2018; Zangrando et al., 2013). These studies show that regional pollutants and local natural aerosol production affect sea ice albedo and the heat balance of the atmosphere, especially in the summer when mid-latitude transport is not as frequent relative to that during the winter/spring Arctic Haze season (Hansen and Nazarenko, 2004; Jacob et al., 2010; Shindell, 2007)."

Line 67-79 " Sea salts, derived from the Arctic Ocean, are the dominant coarse particles (>1 μm) in the Arctic atmosphere (Behrenfeldt et al., 2008; Chi et al., 2015). Compared to other types of aerosols, sea salt is the largest contributor to radiative forcing in remote Ocean air (Wang et al., 2019). Natural sea salt particles can provide large surfaces for heterogeneous reaction with acidic gases in the Arctic air (Chi et al., 2015; Geng et al., 2010; Hara et al., 2003). Moreover, sea salt particles are an important source of cloud condensation nucleation (CCN) in the Arctic air (Abbatt et al., 2019). Coarse dust particles in the Svalbard region have been observed to be occasionally influenced from local (Svalbard) and/or distant (e.g., Iceland, Greenland and Siberia) sources in high latitudes (Behrenfeldt et al., 2008). Tobo et al. (2019)

showed that glacial outwash sediments in Svalbard (a proxy for glacially sourced dusts) due to the recent rapid and widespread retreat of glaciers have a remarkably high ice nucleating ability under conditions relevant for mixed-phase cloud formation."

Line 106-115 " Sulfate is a dominant aerosol component in the Arctic air (Quinn et al., 2007). The Community Earth System Model simulations show that sources from East Asia have the largest contribution to the Arctic sulfate column burden, with an annual mean contribution of 27%, followed by 11–13% each from South Asia, the rest of the world (including the Arctic), and Russia/Belarus/Ukraine sources and 13% from natural sources (Yang et al., 2018). Large amounts of secondary species including sulfate and OM not only change radiative forcing and number of CCN in Arctic atmosphere (Abbatt et al., 2019; Yang et al., 2018) but also influence optical, hygroscopic, and CCN activity of these internally mixed BC and mineral dust particles (Lathem et al., 2013; Raatikainen et al., 2015; Zanatta et al., 2018). "

Comment 2: There is no background on previous relevant studies conducted at the study location, even though there is a long-term monitoring station with aerosol measurements at Ny-Ålesund (https://www.esrl.noaa.gov/psd/iasoa/stations/nyalesund). It is not the same exact location as the Chinese site, but close enough to at least use those routine, publically-available measurements to provide some broader spatial and temporal context. Then, the discussion on the background on sources of BC, OM, BrC, and sulfate and the various mixing mechanisms can reside.

Response: We searched the data portal but there is no aerosol measurement during the year of our observation in the database. The database only has online gaseous concentration (e.g. $CO_2$) during our observation period. We added some background information here.

We added the background as the comments.

line 147-153: "On the west coast of the island of Spitsbergen, Ny-Ålesund is a Norwegian research and monitoring infrastructure, hosting national and international research projects and programmes. The Norwegian Polar Institute (NPI) runs the Sverdrup Research Station at the coast and Zeppelin Observatory at the Mountain 475 m asl, and Sweden, Germany, France, Italy, Japan, China, England, The Netherlands, South Korea, and India are the other countries to have established long-term programmes in Ny-Ålesund (https://www.esrl.noaa.gov/psd/iasoa/stations/nyalesund)."

Comment 3: Laskina et al. (2015) states, "Typically, SSA particles are deposited wet and, if possible, samples used for single-particle analysis should be stored at or near conditions at which they were collected in order to avoid dehydration." I assume the RH was much higher during collection, so can the authors comment on what might change between collection and storage under the different RH conditions?

Response: The TEM and SEM analysis were carried out in vacuum so the particles we analysed are dry at the time of observation. It is a general practice to keep the samples in dry conditions to prevent the sample to be exposed in humid air and phase changes during the storage.

Laskina's paper also mentioned that "…However, if samples need to be dry, as is often the case, then this study found that storing SSA particles at ambient laboratory conditions (17−23% RH and 19−21 °C) was effective at preserving them and reducing changes that would alter samples and subsequent data interpretation."

Comment 4: Also, Laskina et al. is focused on sea spray aerosol, but the authors conclude that continental sources were a major contributor (lines 377-379), so what about loss of semivolatile organics or sulfate during low RH storage? This possible caveats should be discussed in more detail than they currently are.
"Sulfate" particles is used intermittently throughout, perhaps "sulfate-containing" is more appropriate.
Response: This is now mentioned in the text:
Context in section 2.2"In our study, the effects of water and other semi-volatile organics were not considered as they evaporate in the vacuum."

Sulfate does not evaporate so our storage does not have any impact on the sulfate aerosols.

**Specific comments:**
Comment 5: Title: Because this work is only conducted at one location during one month, extending it to "summer Arctic" seems like a stretch as aerosol populations can vary significantly from terrestrial to the high Arctic and can also change from early to late summer. Perhaps the authors should consider changing the title to something like, "Organic coatings on sulfate and soot particles during late summer in the Svalbard Archipelago".
Response: Yes. This is now revised title to " Organic coating on sulfate and soot particles during late summer in the Svalbard Archipelago"

Comment 6: Lines 33-34: State the motivation for the particular focus on these particle types, so that this focus is justified.
Response: We modified the sentence and made clear statement:
Line 32 to 33: determine the size and mixing state of individual sulfate and carbonaceous particles at 100 nm – 2 μm

Comment 7: Lines 34-35: State percentage of OM coated NSS-sulfate here to provide some quantification to "commonly coated".
Response: Revised
Line 33 to 34: found that 74% by number of non-sea salt sulfate particles were coated with organic matter (OM)

Comment 8: Lines 52-53: Regional pollutants and local natural aerosol production can also affect sea ice albedo, especially in the summer when midlatitude transport is not as frequent relative to the winter/spring Arctic Haze season.

Response: Revised

Line 58 to 61: "These studies show that regional pollutants and local natural aerosol production affect sea ice albedo and the heat balance of the atmosphere, especially in the summer when mid-latitude transport is not as frequent relative to that during the winter/spring Arctic Haze season (Hansen and Nazarenko, 2004; Jacob et al., 2010; Shindell, 2007). "

Comment 9: Lines 57-63: This part is very BC focused and should be included in the second paragraph.

Response: Revised. We moved the BC part into second paragraph.

Comment 10: Lines 144-146: There are no details provided on where these meteorological data came from or what instrumentation was used to measure the mentioned parameters. Although, the data are not presented anywhere so perhaps this information is not relevant in include at all.

Response: We added the details about the instrument.

In context: line 166-170 "During the sampling period, meteorological data at the sampling site including pressure (P), relative humidity (RH), temperature (T ), wind speed (WS), and wind direction (WD) were measured and recorded every 5 min using a pocket weather meter (Kestrel 4500, Nielsen-Kellermann Inc., USA)."

Comment 11: Line 193: Define the size of "fine" and quantify "abundant".

Response: We revised them.

Line 215: Here we deleted the fine and add (~30% by number) after the abundant.

Comment 12: Lines 200-201: "good consistent property" is vague.

Response: We revised them.

Section 3.2 line 221-224 " Because all the sulfate particles in different samples collected in the Arctic had similar elemental composition and mixing state from TEM observations, we just selected three samples containing abundant sulfate particles (Table 1) for further studies."

Comment 13: Section 2.4: Separate SEM and AFM into separate sections – not sure why these specifically are combined.

Response: We separated them into two part: section 2.4 SEM measurement and 2.5 AFM measurement.

Comment 14: Line 275: Why only 500 m?

Response: We did 50 m, 500 m, and 1000 m, these three are generally consistent. As a result, we chosen 500 m as a representation. We revised the sentence to:

Section 2.7 line 288-290 "Here we selected an altitude of 500 m as the end point in each back trajectory (Figure 1). Back trajectories for 50 m and 1000 m above sea level are similar."

Comment 15: Lines 276-279: This belongs in the results section.
Response: Thanks. We move this section into result in section 3.2

Comment 16: Lines 290-292: Why were different days evaluated for HYSPLIT versus FLEXPART (3 versus 10 day, respectively)?
Response: These are two different models. HYSPLIT is a meteorological model, tracing the air mass travel to a site. It does not offer conclusive information about the long term transport of particles (Stein et al., 2015). However, the FLEXPART model including WRF model uses a different method to evaluate the particle dispersion along the backward air masses. For each simulation (one per sample), 20000 pseudo-particles were released in a small volume around the station position. Each single particle position evolution backward in time was determined by Lagrangian dispersion calculation. In this study, we based on the data Figure to adjust the period. Here we found that 10 days FLEXPART data can well indicate the particle sources.

As the Figure 3 shown, 3 days back trajectories provided sufficient information to indicate the air masses movement. Furthermore, long-term time back trajectories become less certain.

Reference: Stein A. F., et al. NOAA's Hysplit Atmospheric Transport and Dispersion Modeling System. Bull. Amer. Meteorol. Soc. 96, 2059-2077 (2015).
Brioude, J., D. Arnold, A. Stohl, M. Cassiani, D. Morton, P. Seibert, W. Angevine, S. Evan, A. Dingwell, J. D. Fast, R. C. Easter, I. Pisso, J. Burkhart, and G. Wotawa (2013), The Lagrangian particle dispersion model FLEXPART-WRF version 3.1, Geosci. Model Dev., 6(6), 1889-1904.

Comment 17: Lines 304-306: Provide a citation for this statement.
Response: added two reference here.
Moffet, R. C., T. C. Rödel, S. T. Kelly, X. Y. Yu, G. T. Carroll, J. Fast, R. A. Zaveri, A. Laskin, and M. K. Gilles (2013), Spectro-microscopic measurements of carbonaceous aerosol aging in Central California, Atmos. Chem. Phys., 13(20), 10445-10459.
Li, W., J. Sun, L. Xu, Z. Shi, N. Riemer, Y. Sun, P. Fu, J. Zhang, Y. Lin, X. Wang, L. Shao, J. Chen, X. Zhang, Z. Wang, and W. Wang (2016), A conceptual framework for mixing structures in individual aerosol particles, J. Geophys. Res., 121(22), 13,784-713,798.
Riemer, N., A. P. Ault, M. West, R. L. Craig, and J. H. Curtis (2019), Aerosol Mixing State: Measurements, Modeling, and Impacts, Rev. Geophys., 57, https://doi.org/10.1029/2018RG000615.

Comment 18: Lines 310-312: 39% for the S-rich + soot + OM out of all NSS-particles? How was NSS-sulfate and OM determined?
Response: We reworded this sentence to :
Section 3.1 line 322-325 "Here, we focused on non-sea salt (NSS) particles including S-rich, soot, and OM particles NSS-particles and sulfate-containing particles account for 39±5% and 29±7% by number of all the 2002 particles analysed (Figure 3)."

Comment 19: Lines 332-333: How was this documented? This statement is a bit vague.
Response: We deleted it

Comment 20: Figure 1: The date are very small and it is difficult to tell if there is any sort of
time evolution in the air mass sources throughout August. Perhaps the authors could either
color the lines by date or create multiple panels (e.g., 1 per week of trajectories).
Response: We revised the Figure as below. The one week is not good at all. The current data
can know where these air masses were from.

[Figure]

**Figure 1** 72 h back trajectories of air masses at 500 m over Arctic Yellow River Station in Svalbard during 3–26 August 2012, and arriving time was set according to the sampling time. Air masses were divided into two groups by the yellow line: one group from the central Arctic Ocean and the other one from North America and Greenland. Pie charts show the number fractions of sea salt, S-rich, and other particles.

Comment 21: Figure 3: I assume this is from TEM/EDX? Why are soot and OM not shown?
Also, can the authors show a time series in addition to the diameter relative fraction? That
might provide some insight into temporal changes due to air mass origin.
Response: We revised the Figure. More than 98% of soot and OM were internally mixed sulfate, named as sulfate-coating particles (Figure 7). The very small number of soot and OM and some mineral particles were assigned into others.

As the reviewer's comments, air masses were divided into two groups by the yellow line: one group from the central Arctic Ocean and the other one from North America and Greenland. Pie charts show the number fractions of sea salt, S-rich, and others. Here we can find that more S-rich particles were from North America and more sea salt particles from ocean area (Figure 1).

Comment 22: Table S1: There is a substantial amount of referencing to Table S1. While I am glad to see the authors have include sample and particle numbers for each analytical technique, jumping back and forth between the manuscript and SI is tedious. I suggest the table be moved to the main manuscript.
Response: Thanks. We moved to the main manuscript.

**Technical corrections:**
Comment 23: There are many typos and grammatical issues that need to be fixed throughout the manuscript.
Line 32: "mixing state properties"
Response: We smooth the English writing in all the manuscript.

[revised manuscript text omitted]